# Crosstalk between AML and stromal cells triggers acetate secretion through the metabolic rewiring of stromal cells

Nuria Vilaplana-Lopera[1], Vincent Cuminetti[2†], Ruba Almaghrabi[1,3†], Grigorios Papatzikas[1,4†], Ashok Kumar Rout[5], Mark Jeeves[1], Elena González[1], Yara Alyahyawi[1,6], Alan Cunningham[7], Ayşegül Erdem[7], Frank Schnütgen[8,9,10], Manoj Raghavan[1,11], Sandeep Potluri[1,11], Jean-Baptiste Cazier[1,4], Jan Jacob Schuringa[7], Michelle AC Reed[1], Lorena Arranz[2], Ulrich L Günther[1,5*‡], Paloma Garcia[1*‡]

[1]Institute of Cancer and Genomic Sciences, University of Birmingham, Birmingham, United Kingdom; [2]Stem Cells, Ageing and Cancer Research Group, Department of Medical Biology, Faculty of Health Sciences, UiT – The Arctic University of Norway, Tromso, Norway; [3]Department of Laboratory Medicine (hematology), Faculty of Applied Medical Sciences. Albaha University, Kingdom of Saudi Arabia, Al Bahah, Saudi Arabia; [4]Centre for Computational Biology, University of Birmingham, Birmingham, United Kingdom; [5]Institute of Chemistry and Metabolomics, University of Lübeck, Lübeck, Germany; [6]Department of Medical Laboratories Technology, College of Applied Medical Sciences, Jazan University, Jazan, Saudi Arabia; [7]Department of Experimental Hematology, University Medical Center Groningen, University of Groningen, Groningen, Netherlands; [8]Department of Medicine, Hematology/Oncology, University Hospital Frankfurt, Goethe University Frankfurt, Frankfurt, Germany; [9]Frankfurt Cancer Institute, Goethe University Frankfurt, Frankfurt, Germany; [10]German Cancer Consortium (DKTK), partner site Frankfurt/Mainz, and German Cancer Research Center (DKFZ), Heidelberg, Germany; [11]Centre for Clinical Haematology, University Hospitals Birmingham NHS Foundation Trust, Queen Elizabeth Hospital, Queen Elizabeth Medical Centre, Birmingham, United Kingdom

*For correspondence:
ulrich.guenther@uni-luebeck.de
(ULG);
p.garcia@bham.ac.uk (PG)

†These authors contributed equally to this work

‡Co-senior author

**Competing interest:** The authors declare that no competing interests exist.

**Abstract** Acute myeloid leukaemia (AML) cells interact and modulate components of their surrounding microenvironment into their own benefit. Stromal cells have been shown to support AML survival and progression through various mechanisms. Nonetheless, whether AML cells could establish beneficial metabolic interactions with stromal cells is underexplored. By using a combination of human AML cell lines and AML patient samples together with mouse stromal cells and a MLL-AF9 mouse model, here we identify a novel metabolic crosstalk between AML and stromal cells where AML cells prompt stromal cells to secrete acetate for their own consumption to feed the tricarboxylic acid cycle (TCA) and lipid biosynthesis. By performing transcriptome analysis and tracer-based metabolic NMR analysis, we observe that stromal cells present a higher rate of glycolysis when co-cultured with AML cells. We also find that acetate in stromal cells is derived from pyruvate via chemical conversion under the influence of reactive oxygen species (ROS) following ROS transfer from AML to stromal cells via gap junctions. Overall, we present a unique metabolic communication between AML and stromal cells and propose two different molecular targets, ACSS2 and gap junctions, that could potentially be exploited for adjuvant therapy.

## Editor's evaluation

This article will be of interest to those working in the fields of hematopoiesis, leukemia and cancer microenvironment. The work describes a novel phenomenon whereby the direct interaction of acute myeloid leukemia (AML) cell lines and bone marrow stromal cell lines results in increased production of extracellular acetate in stromal cells, which can then be metabolised by the AML cells. Overall, the data are very thought-provoking but future work some of which is technically challenging, will be necessary for a full appreciation of the biological relevance of this interaction.

## Introduction

Acute myeloid leukaemia (AML) is a heterogeneous multiclonal disease characterised by a rapid proliferation of aberrant immature myeloid cells that accumulate in the bone marrow, and eventually in the blood and other organs, severely impairing normal haematopoiesis. AML cells show a highly adaptive metabolism that allows them to efficiently use a variety of nutrients to obtain energy and generate biomass (reviewed in *Kreitz et al., 2019*). This high metabolic plasticity confers AML cells a strong advantage against normal haematopoietic cells and has been related to AML aggressiveness. Although the metabolism in AML cells has been broadly investigated (*Kreitz et al., 2019*), fewer studies have focused on identifying metabolic alterations related to the interaction between AML and niche cells. For instance, AML cells are known to interact and modulate niche components for their own support by secreting soluble factors (*Schelker et al., 2018*; *Passaro et al., 2017*; *Zeng et al., 2006*; *Carey et al., 2017*; *Zhang et al., 2020*), via exosomes (*Wang et al., 2019*; *Kumar et al., 2018*; *Hornick et al., 2016*) or by establishing direct interactions, mediated by gap junctions (*Kouzi et al., 2020*) or tunnelling nanotubes (*Omsland et al., 2017*). These interactions with components of the niche provide AML cells with survival cues and chemoresistance, ultimately contributing to increased relapse in AML patients (*Schelker et al., 2018*; *Zeng et al., 2006*; *Wang et al., 2019*; *Kouzi et al., 2020*; *Ye et al., 2016*; *Moschoi et al., 2016*; *Forte et al., 2020*). Furthermore, it has been reported that adipocytes in the niche secrete fatty acids, which are metabolised by AML cells through β-oxidation to obtain energy, protecting AML cells from apoptosis and ROS (*Shafat et al., 2017*; *Tabe et al., 2017*).

As a consequence of their highly proliferative demand, AML cells (*Baccelli et al., 2019*; *Pollyea et al., 2018*; *Molina et al., 2018*; *Lagadinou et al., 2013*) and, particularly, chemotherapy-resistant AML cells (*Farge et al., 2017*) present abnormally high levels of reactive oxygen species (ROS) (*Li et al., 2011*; *Hole et al., 2013*). How AML cells cope with high ROS levels has been intriguing and recent reports are shedding some light on whether the microenvironment plays a role in the redox metabolism of AML cells. For instance, it was reported that Nestin[+] bone marrow mesenchymal stem cells (BMSCs) support AML progression by increasing the bioenergetic capacity of AML cells and providing them with glutathione (GSH)-mediated antioxidant defence to balance the excess ROS (*Forte et al., 2020*). Similarly, a recent study showed that co-culturing BMSCs with AML cells leads to a decrease in AML ROS levels due to an activation of the antioxidant enzyme GPx-3 in AML cells (*Vignon et al., 2020*).

Our work provides new insight into the metabolic and redox crosstalk between AML and stromal cells, revealing a new metabolic interaction between AML and stromal cells. By combining transcriptomic and nuclear magnetic resonance (NMR) data, our results demonstrate that stromal cell metabolism is rewired in co-culture resulting in higher glycolysis and pyruvate decarboxylation, leading to acetate secretion. Our results also show that AML cells are able to transfer ROS to stromal cells by direct interaction through gap junctions and that these ROS can be used by stromal cells to generate and secrete acetate, which is utilised by AML cells to feed the TCA cycle and to generate lipids. Targeting ROS transfer via modulation of gap junctions to suppress acetate provision by stromal cells or targeting acetate usage could serve as an adjuvant therapy to eradicate AML.

## Results

### Co-culturing AML and stromal cells in direct contact triggers acetate secretion by stromal cells

We first sought to determine whether interactions between AML and stromal cells in co-culture would result in differences in the consumption or production of extracellular metabolites. For this purpose, three human AML cell lines (SKM-1, Kasumi-1 and HL-60) representing different AML subtypes (M5, M2 t(8;21), and M2, respectively) were cultured separately and in co-culture with MS-5, a stromal mouse cell line capable of maintaining haematopoiesis (*Itoh et al., 1989*). The metabolic composition of the extracellular medium in each condition was analysed by [1]H-NMR and compared at different time points. The most striking difference found in co-culture compared to cells cultured separately was an increased secretion of acetate, which was common for the three cell lines used (*Figure 1A and B*). Moreover, only stromal cells secreted acetate to a lower extent when cultured alone whereas AML cells did not secrete any acetate when cultured alone. Altogether, these findings suggest that acetate secretion is a result of a direct interaction between AML and stromal cells. In addition, the observation that only stromal cells secrete acetate under these conditions suggests that stromal cells could be responsible for the increased acetate secretion found in co-culture.

We further examined the levels of other common extracellular metabolites, including glucose, lactate, glutamate and glutamine. As shown in *Figure 1—figure supplement 1*, higher consumption of glucose along with a higher secretion of lactate was observed in AML and stromal cells in co-culture compared to single cultures, suggesting a higher glycolytic flux in co-culture. However, the levels of glucose consumption and lactate production in co-culture were similar to the sum of the glucose consumption and lactate production levels of the AML and stromal cells in single cultures, suggesting that the overall increase in glycolysis in co-culture was just a result of culturing both cell types together. Additionally, we observed no variation in glutamate and glutamine levels suggesting that these metabolites are not involved in interactions that result from co-culture and are utilised depending on their availability.

The metabolic differences found in co-culture, could be due to altered proliferation under these conditions. To determine whether this was the case, a CFSE-based proliferation assay was performed in which AML cellsbut not stromal cells were stained and their growth compared when cultured separately vs in co-culture. Over 48 hr, none of the human AML cell lines tested presented differences in their proliferation rates (*Figure 1—figure supplement 2A*), thus confirming that the metabolic changes found in co-culture were caused by a mechanism independent of changes in proliferation.

Next, we aimed to determine whether cell-to-cell contact could play a role in the increased acetate secretion found in co-culture. For this we co-cultured AML and stromal cells separated by a permeable membrane, allowing cells to share the extracellular medium but impeding cell-to-cell contact. Co-culturing cells using a permeable membrane blocked the increase in acetate secretion observed under direct contact conditions (*Figure 1C*). In fact, cells in co-culture presented lower levels of acetate than MS-5 cells cultured alone revealing that direct cell-to-cell contact is required for acetate secretion in co-culture.

To examine whether increased acetate secretion is specific for the interaction of AML cells with stromal cells, we co-cultured the human AML cell lines with the cervical human cancer cell line HeLa and compared the levels of acetate of each cell type cultured alone. We found that there was no acetate secretion in co-culture, also not by HeLa and AML cells cultured alone (*Figure 1—figure supplement 2B*), suggesting that increased acetate secretion is specific for an AML-stromal cell interaction.

Considering that our previous data seemed to indicate that MS-5 cells were responsible for acetate secretion when co-cultured with human AML cell lines, we decided to investigate how the levels of extracellular acetate would vary after separating cells from co-culture. The three human AML cell lines were co-cultured for 24 hr with the MS-5 mouse stromal cells prior to separation and subsequent culture in the same spent media. Extracellular acetate levels in previously co-cultured MS-5 cells followed a similar trend as before separation, suggesting that the MS-5 cells most likely are responsible for the increased acetate secretion found in co-culture (*Figure 1D*). Moreover, AML cells did not follow this trend as they either maintained the levels of acetate seen prior to separating the cells (Kasumi-1 and HL-60) or presented only a moderate increase (SKM-1) after being separated from co-culture.

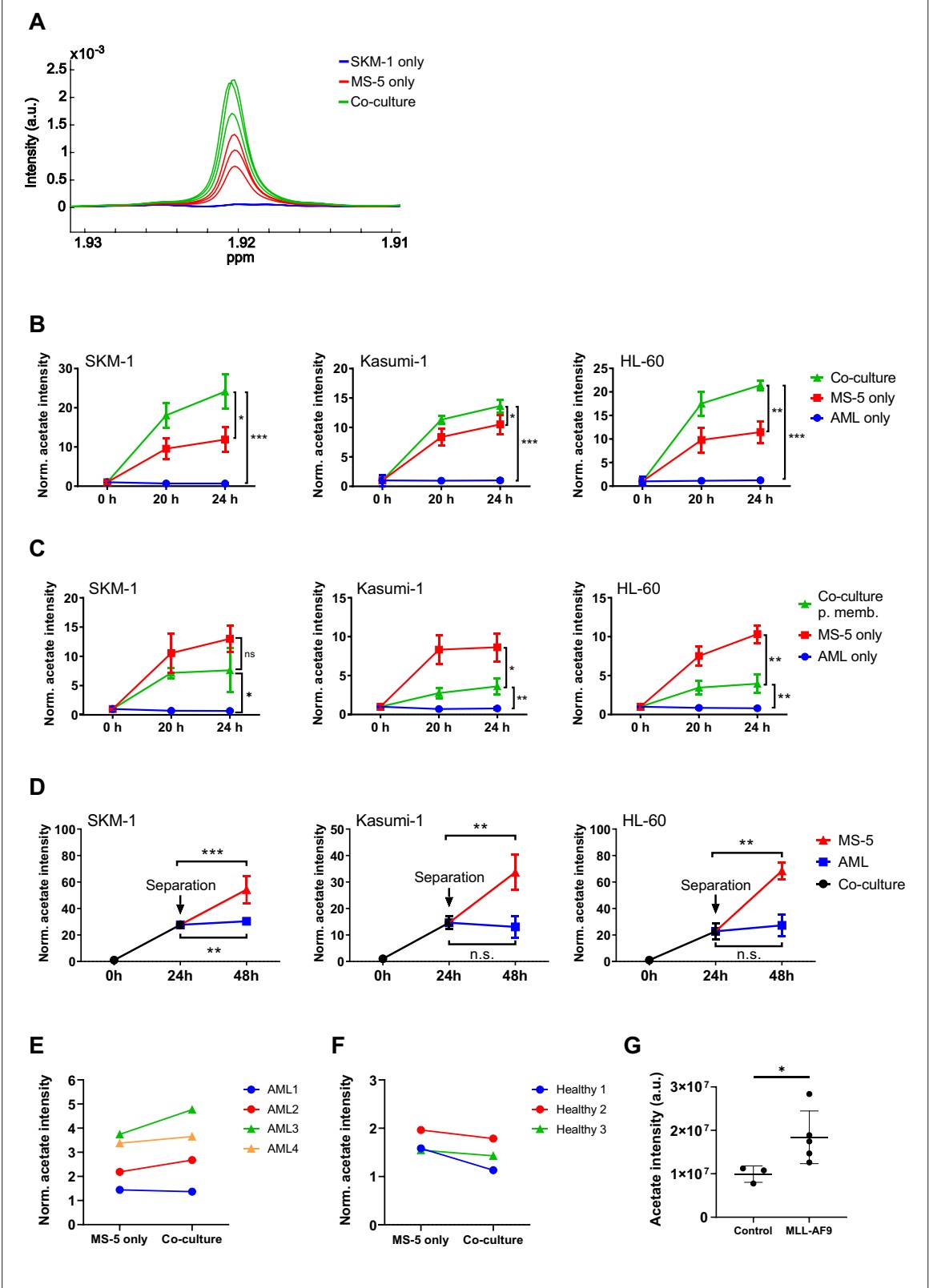

**Figure 1.** Acetate secretion by stromal cells increases in AML-stroma co-cultures of several AML cell lines and primary AML cells in direct contact. (**A**) Section of $^1$H-NMR spectra, corresponding to the methyl group of acetate, from extracellular medium samples of SKM-1 cells cultured alone (blue), MS-5 cells cultured alone (red) and SKM-1 and MS-5 cells in co-culture (green) after 24 hours. (**B**) Extracellular acetate levels in AML cell lines (SKM-1, Kasumi-1 and HL-60) cultured alone (blue), MS-5 cells cultured alone (red) and AML and MS-5 cells in co-culture in direct contact (green) at 0, 20 and

*Figure 1 continued on next page*

*Figure 1 continued*

24 hours of incubation. Each point represents the mean of n=3 independent experiments and error bars represent standard deviations. (**C**) Extracellular acetate levels in AML cell lines cultured alone (blue), MS-5 cells cultured alone (red) and AML and MS-5 cells in co-culture separated by a 0.4 µm permeable membrane (green) at 0, 20, and 24 hr of incubation. Each point represents the mean of n=3 independent experiments and error bars represent standard deviations. (**D**) Extracellular acetate levels in AML cell lines and MS-5 cells in co-culture (black) for 24 hr and after being separated and cultured alone in the same medium until 48 hr (blue for AML and red for MS-5). Each point represents the mean of n=3 independent experiments and error bars represent standard deviations. (**E**) Extracellular acetate levels in MS-5 cells cultured alone and primary patient-derived AML cells co-cultured with MS-5 cells at 48 hr. Each set of points represents an independent experiment (n=4). (**F**) Extracellular acetate levels in MS-5 cells cultured alone and healthy donor-derived peripheral blood mononuclear CD34+ (PBMC) cells co-cultured with MS-5 cells at 48 hr. Each set of points represents an independent experiment (n=3). For **E** and **F**, symbols (circles or triangles) indicate same cell culture medium composition was used. (**G**), Acetate levels in bone marrow extracellular fluid of C57BL6/J mice 6 months after transplantation with bone marrow nucleated cells isolated from control or MLL-AF9 transgenic mice. For B, C, and D unpaired Student's t-tests were applied for each condition (black brackets) for **G** a Mann-Whitney test was applied (black brackets). p-values are represented by n.s. for not significant * for p-value <0.05, ** for p-value <0.01 and *** for p-value <0.001.

The online version of this article includes the following source data and figure supplement(s) for figure 1:

**Source data 1.** Values and stats for panels included in *Figure 1*.

**Figure supplement 1.** Other extracellular metabolite levels in co-cultures with SKM-1 and MS-5 cells.

**Figure supplement 2.** Proliferation in co-culture, co-culturing AML cells with an unrelated cell line (HeLa).

**Figure supplement 2—source data 1.** Values obtained for cell proliferation with CFSE in AML cell lines cultured alone vs in coculture (A) and raw extracellular acetate values obtained for SKM-1 grown in cocultured with HeLa cells.

We further investigated whether increased acetate secretion could take place in primary co-cultures using AML cells derived from patients and whether acetate secretion in co-culture could be specific for AML cells. To address this question, we isolated the CD34+ population of four primary AML patient samples, and of three independent healthy donors (*Supplementary file 1*), cultured them in co-culture with MS-5 cells, and analysed the composition of the extracellular medium after 48 hr. We found that three out of four primary AML samples presented higher levels of acetate when co-cultured with MS-5 cells compared to MS-5 cells cultured alone (*Figure 1E*). Contrary, none of the healthy donor samples showed an increased acetate secretion in co-culture suggesting that acetate secretion is specific for AML cells in co-culture (*Figure 1F*).

Furthermore, we sought to determine whether in an in vivo setting, increased acetate production would be observed. For this, acetate levels were analysed in the bone marrow extracellular fluid (BMEF) of mice transplanted either with mouse MLL-AF9+ leukaemic cells or with healthy wild type mouse hematopoietic cells. These experiments revealed that a significantly higher amount of acetate was present in the BMEF of mice suffering from leukaemia compared to controls (*Figure 1G*).

## AML cells consume and use acetate secreted by stromal cells to feed the TCA cycle and for lipogenesis

Following the finding that stromal cells might be responsible for acetate secretion in co-culture, we next examined whether AML cells could metabolise the secreted acetate. We first sought to define the concentration of secreted acetate in the extracellular medium in co-culture. For this, we compared a sample of extracellular medium from a co-culture of SKM-1 and MS-5 cells after 24 hr to a calibration curve (*Figure 2—figure supplement 1A*), which allowed us to determine the concentration of acetate in co-culture as approximately 3–4 mM. We then investigated whether SKM-1 cells can consume acetate both in normal plasma concentrations (*Gao et al., 2016*) and co-culture concentrations (0.25 mM or 3 mM, respectively). In both cases, SKM-1 cells consumed acetate significantly after 48 hr, and in normal plasma conditions also after 24 hr (*Figure 2—figure supplement 1B*).

We then employed a tracer-based approach using [2-$^{13}$C]acetate to assess whether SKM-1 or MS-5 cells can utilise acetate (*Figure 2A* and *Figure 2—figure supplement 2A*). Both cell types imported $^{13}$C labelled acetate as observed in NMR spectra (*Figure 2B*). However, only SKM-1 cells showed $^{13}$C label incorporation in several TCA cycle related metabolites, including aspartate, citrate, glutamate, 2-oxoglutarate, glutathione, and proline, as well as in acetylcarnitine. Acetylcarnitine is known to be produced by cells when large amounts of acetyl-CoA are present in the mitochondria (*Childress et al., 1967*; *Stephens et al., 2007*), which could be in line with this experiment in which a high concentration of [2-$^{13}$C]acetate (4 mM) was used. Overall, this data suggests that acetate in co-culture could be utilised by SKM-1 cells but not by MS-5 cells.

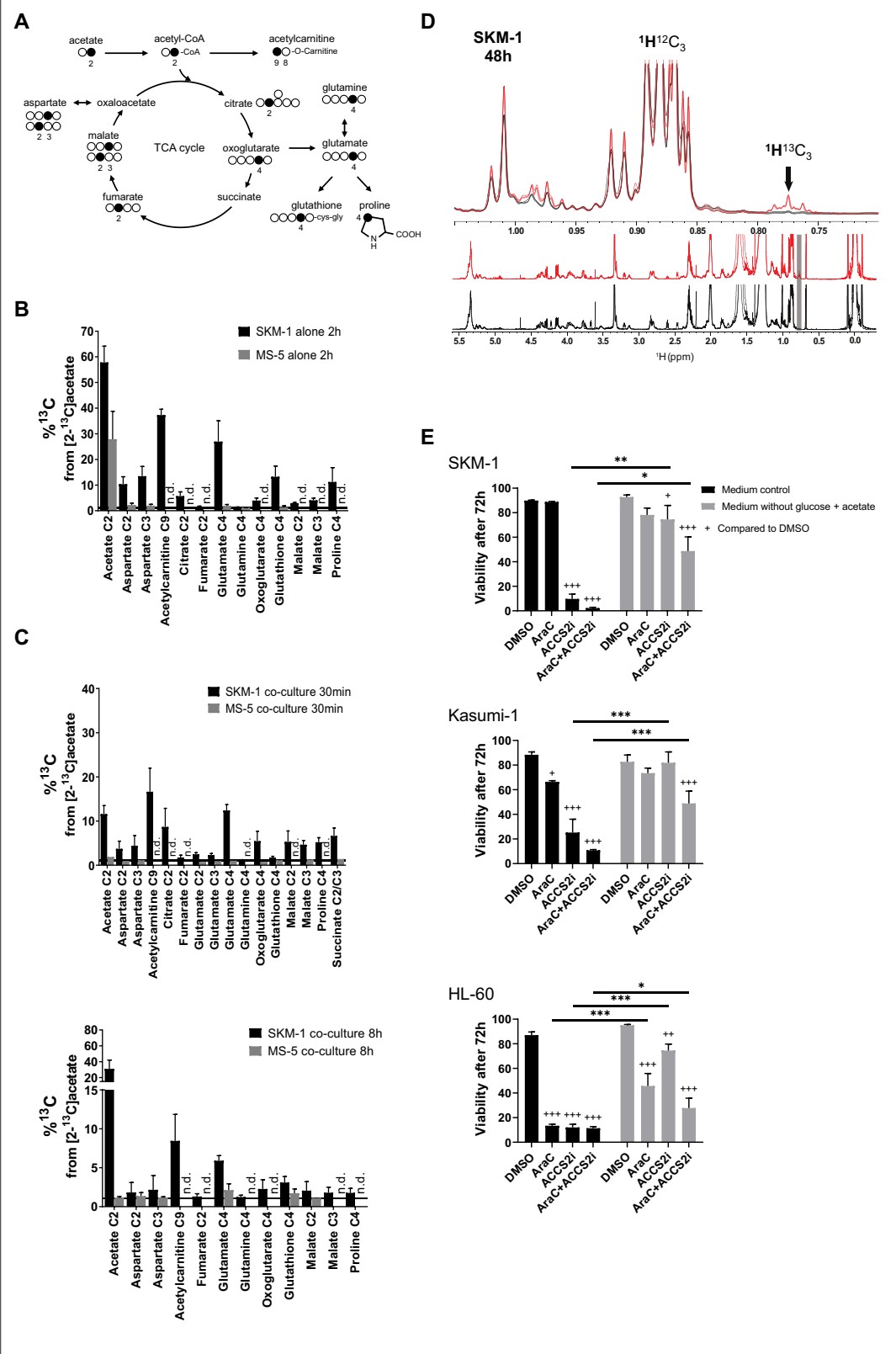

**Figure 2.** AML cells can import the secreted acetate in co-culture to use it in TCA cycle and lipid biogenesis. (**A**) Schematic of label distribution arising from [2-$^{13}$C]acetate in TCA cycle intermediates. Black circles correspond to positions expected to be labelled. (**B**) $^{13}$C percentages of label incorporation in polar metabolites from labelled acetate in SKM-1 and MS-5 cells after two hours of incubation with [2-$^{13}$C]acetate (4 mM) derived from $^1$H-$^{13}$C-

*Figure 2 continued on next page*

*Figure 2 continued*

HSQC NMR spectra. Bars represent the mean of the $^{13}$C percentages and error bars represent the standard deviations for n=3 independent experiments. (**C**) $^{13}$C percentages on polar metabolites in SKM-1 and MS-5 cells in co-culture. Cells were co-cultured for 24 hr before the addition of extra 4 mM sodium [2-$^{13}$C]acetate and culture for additional 30 min (upper panel) or 8 hr (lower panel). Bars represent the mean of the $^{13}$C percentages and error bars represent the standard deviations of n=3 independent experiments. $^{13}$C natural abundance is represented as a black bar at %$^{13}$C=1.1. (**D**) $^{1}$H 1D NMR of lipids extracted from SKM-1 cells. Cells were co-cultured with MS-5 cells for 24 hr before the addition of extra 4 mM sodium [2-$^{13}$C]acetate and culture for additional 48 hr. Lower panel represents overlay of spectra (n=3) from cells grown in $^{12}$C- and $^{13}$C—labelled acetate (black and red, respectively). Upper panel is the zoomed section of the spectra show $^{1}$H$^{13}$C-methyl signal multiplets at 1.05ppm to 0.7pm as indicated by an arrow (the shift of the $^{1}$H$^{13}$C methyl satellite signal is caused by the scalar $J_{CH}$ coupling of 125–128 Hz). (**E**) Cell viability measured by propidium iodide staining after culturing for 72 hr in glucose-free media containing 4 mM acetate or normal media, in the presence of DMSO, AraC 1 μM, and ACSS2i 20 uM. Bars represent the mean and error bars represent the standard deviations for n=3 independent experiments. A Tukey's multiple comparison test was performed comparing each treatment and different medium conditions and p-values are represented by n.s. for not significant * for p-value <0.05, ** for p-value <0.01 and *** for p-value <0.001.

The online version of this article includes the following source data and figure supplement(s) for figure 2:

**Source data 1.** Data and stats for panels included in *Figure 2*.

**Figure supplement 1.** Titration of acetate concentration in co-culture and acetate consumption by AML cells.

**Figure supplement 1—source data 1.** Raw values for acetate titration (A) and acetate consumption by SKM1 (B).

**Figure supplement 2.** Label incorporation from [2-$^{13}$C]acetate in SKM-1 and MS-5 cells cultured alone.

**Figure supplement 2—source data 1.** Raw values for different metabolites showing label incorporation from acetate in coculture 30 min incubation.

**Figure supplement 3.** Acetate label incorporation analysis in lipids in AML cells in co-culture after 8 hr of labelling.

**Figure supplement 4.** Acetate label incorporation analysis in lipids in AML cells in co-culture after 48 hr of labelling.

**Figure supplement 5.** Acetate label incorporation in TCA metabolites after 24 hr of ACSS2i treatment.

**Figure supplement 5—source data 1.** Raw values of acetate labelling in SKM-1 cells +/-ACSS2 i.

To determine whether AML cells can import and metabolise the secreted acetate in co-culture, we co-cultured AML and MS-5 cells for 24 hr and then added [2-$^{13}$C]acetate to the spent extracellular medium and cultured cells for 30 min and 8 hr before analysing the intracellular metabolites in each cell type (*Figure 2C*). We found that only AML cells incorporated $^{13}$C into intracellular metabolites at both timepoints (*Figure 2C* and *Figure 2—figure supplement 2B-C*). The intake of acetate by AML cells was very rapid, with several TCA cycle metabolites such as glutamate, oxoglutarate, malate, proline, and succinate showing label incorporation after 30 min of [2-$^{13}$C]acetate labelling. The metabolisation pattern for all AML cells in co-culture was similar to that observed for SKM-1 cells alone (*Figure 2B and C* and *Figure 2—figure supplement 2B-C*). We also investigated whether the acetate taken by AML cells undergoes alternative fates other than TCA cycle. For this purpose, we co-cultured AML and MS-5 cells for 24 hr before adding [2-$^{13}$C]acetate to the spent extracellular medium and cultured them for further 8 or 48 hr before analysing the intracellular metabolites. No $^{13}$C labelling was incorporated into lipids when co-cultures were labelled with [2-$^{13}$C]acetate for 8 hr (*Figure 2—figure supplement 3*). In contrast, at 48 hr the three AML cell lines presented $^{1}$H$^{13}$C-methyl signals at 0.7–0.8 ppm confirming labelling of lipid CH$_3$ groups (approximately 10–20% enrichment) (*Figure 2D* and *Figure 2—figure supplement 4*).

Overall, these results indicated that AML cells could uptake and utilise the acetate secreted by stromal cells in co-culture as a substrate to feed into the TCA cycle and for lipid biosynthesis.

To understand the possible physiological relevance of our findings, we cultured the three human AML cell lines in normal media and glucose-free media in the presence of acetate, and treated them or not with the chemotherapy agent cytarabine (cytosine arabinoside, AraC). Our data showed that the AML cells studied displayed different degrees of sensitivity to AraC, with HL-60 cells being more sensitive and Kasumi-1 more resistant (*Figure 2E*). Interestingly, we observed that the high sensitivity for AraC displayed by HL-60 cells could be partially counteracted by the presence of acetate in the cultures. We then investigated whether the inhibition of the acetyl CoA synthetase short-chain family

member 2 (ACSS2i), has an effect of the survival of the cells (*Figure 2E*). All human AML cell lines studied displayed high sensitivity to the ACSS2i when cultured in glucose with no added acetate, indicating the importance of ACSS2 in metabolism, as previously observed in myeloma cells (*Li et al., 2021*). In glucose-free media supplemented with 4 mM acetate, the survival of the cells was maintained. Interestingly, under this condition, the ACSS2i treatment alone did not have a profound detrimental effect on the survival of the cells. By performing [2-$^{13}$C]acetate label incorporation we observed that the ACSS2i reduced the incorporation of acetate to the TCA metabolites by approximately 60% after 24 hr of treatment (*Figure 2—figure supplement 5*), indicating the incomplete inhibition of the enzyme and thus still accessibility of acetate usage by the cell, providing an explanation for the higher survival in the medium supplemented with acetate compared to control media. Nonetheless, it was clear that when cells were treated with both drugs, ACSS2i treatment sensitised AML cells to AraC treatment. Our results suggest that acetate confers chemoresistance to AML cells and that the use of ACSS2i sensitizes AML cells to conventional chemotherapies such as AraC.

## Transcriptomic data highlights a metabolic rewiring of stromal cells in co-culture characterised by upregulation of glycolysis and downregulation of pyruvate dehydrogenase

After establishing that stromal cells are responsible for acetate secretion (*Figure 1D*) and that AML cells consumed the secreted acetate in co-culture (*Figure 2C* and *Figure 2—figure supplement 2*), we sought to elucidate the mechanism behind acetate secretion by MS-5 cells in co-culture. Thus, we set out to perform global gene expression profiling by RNA-seq, comparing cells cultured alone and in co-culture (*Figure 3—figure supplement 1A*). With this approach we identified 587 differentially expressed genes (q-value <0.1) (*Figure 3—figure supplement 1B*); with 476 genes being upregulated and 111 genes being downregulated in co-culture.

Following clustering of differentially expressed genes, gene set enrichment analysis (GSEA) was performed (*Figure 3A*) revealing a positive correlation with the expression of genes that are part of the glycolysis pathway (MSigDB: M5937), as well as the reactive oxygen species pathway (MsigDB: M5938) (*Figure 3B*).

A closer examination of the genes involved in the glycolysis pathway revealed a major upregulation of several glycolysis-related genes in MS-5 cells in co-culture (*Figure 3C* and *Figure 3—figure supplement 1C*). Most of the genes involved in glucose transport (*Slc2a1* and *Slc2a4*) and glucose breakdown to pyruvate (*Pgm1*, *Hk2*, *Gpi1*, *Pfkl*, *Gapdh*, *Pgk1*, *Pgam1*, *Pgam2*, *Eno1*, *Eno1b*, *Eno2*, and *Pkm*) were upregulated, including the gene encoding for the 6-phosphofructo-2-kinase/fructose-2,6-biphosphatase 3 (*Pfkfb3*), a well-known activator of glycolysis.

In contrast, individual examination of genes related to pyruvate metabolism revealed that the genes related to pyruvate conversion to acetyl-CoA were downregulated (*Figure 3C* and *Figure 3—figure supplement 1C*). Pyruvate transport into the mitochondria (*Mpc1*, *Mpc2*) and several pyruvate dehydrogenase (PDH) complex-related genes (*Pdha1*, *Pdhb*, and *Pdhx*), involved in the conversion of pyruvate to acetyl-CoA, remained largely unaltered or were slightly downregulated by co-culture. Additionally, the pyruvate dehydrogenase kinases (*Pdk1*, *Pdk2*, and *Pdk4*), which inhibit the activity of the PDH complex, were found to be upregulated by co-culture (*Figure 3C* and *Figure 3—figure supplement 1C*). Interestingly, the acetyl-CoA synthetases (ACSs), which can generate acetyl-CoA from acetate but have also been reported to perform the reverse reaction, encoded by *Acss2* and *Acss3*, were found to be slightly upregulated in co-culture. Pyruvate can also be converted to 2-oxaloacetate via pyruvate carboxylase (*Pcx*), to lactate by lactate dehydrogenase (*Ldha*), and to alanine via alanine transaminase (*Gpt*). Only *Ldha* was found to be moderately upregulated in co-culture. However, we did not observe a substantial increase in lactate production experimentally in co-culture. Altogether, transcriptomic data suggests that MS-5 cells in co-culture present a major upregulation of glycolysis and downregulation of the PDH complex.

To explore whether glycolysis is upregulated in MS-5 cells in co-culture at the metabolic level and whether acetate could derive from glucose, we performed [U-$^{13}$C]glucose tracing on AML and MS-5 cells cultured alone and in co-culture and analysed the label incorporation in glycolysis-related extracellular metabolites (*Figure 3D*). For the three human AML cell lines tested, extracellular acetate presented significantly higher label incorporations from [U-$^{13}$C]glucose in co-culture compared to cell types cultured alone, providing evidence that the secreted acetate in co-culture derives from glucose.

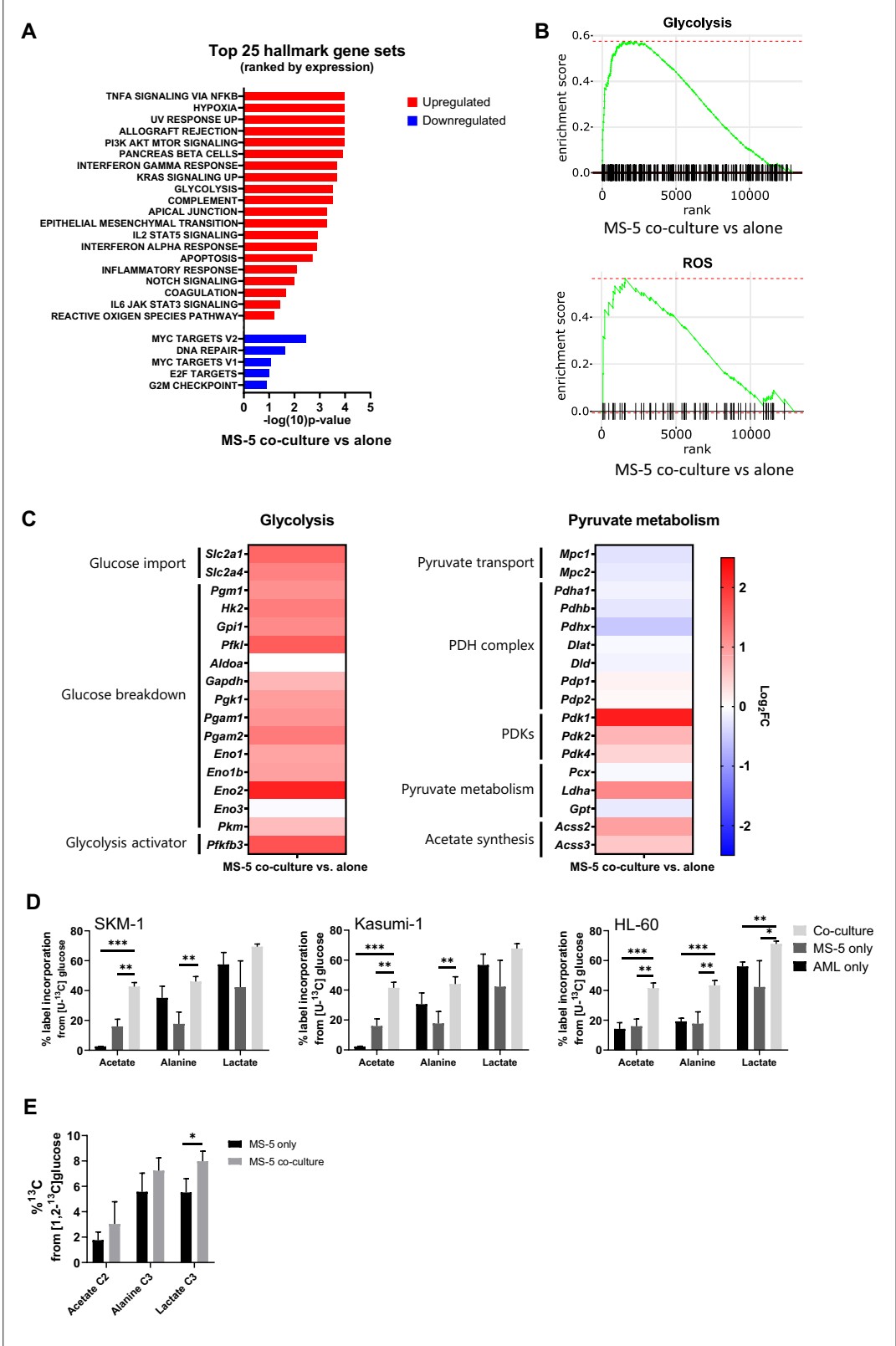

**Figure 3.** Transcriptomic data and [U-$^{13}$C]glucose labelling reveal that stromal cell metabolism is shifted towards higher glycolysis and ROS upon co-culture with AML cells. (**A**) Top 25 GSEA hallmark gene sets ranked by expression in MS-5 cells only vs co-culture with SKM-1 cells, analysed using the collection of hallmark gene sets from Molecular Signature Database with a false discovery rate threshold at 5%. p-Values for each pathway are

*Figure 3 continued on next page*

*Figure 3 continued*

presented as –log(10)p-value. (**B**) GSEA enrichment score plots of glycolysis and ROS generated using Sleuth 0.30.0 R statistical package. (**C**) Fold change values of detected gene transcripts (TPMs) related to glycolysis and pyruvate metabolism. FC values are represented as $\log_2$FC, red values indicate upregulation and blue values indicate downregulation in MS-5 cells in co-culture. (**D**) Label incorporation from [U-$^{13}$C]glucose into extracellular metabolites in AML and MS-5 cells cultured alone or in co-culture after 24 hr. Bars represent the mean of n=3 independent experiments and error bars represent standard deviations. (**E**), Label incorporation from [U-$^{13}$C] glucose into intracellular acetate, alanine and lactate in MS-5 cells cultured alone or in co-culture after 24 hr. Bars represent the mean of n=3 independent experiments and error bars represent standard deviations. p-Values are represented by n.s. for not significant * for p-value <0.05, ** for p-value <0.01 and *** for p-value <0.001.

The online version of this article includes the following source data and figure supplement(s) for figure 3:

**Source data 1.** Data and stats for panels included in *Figure 3*.

**Figure supplement 1.** PCA component analysis, heat map of differentially expressed genes and qPCR for MS-5 cells cultured alone and in co-culture with SKM-1 cells.

**Figure supplement 1—source data 1.** mRNA expression in MS-5 cells cocultured with SKM-1 cells relative to MS-5 alone.

Lactate and alanine, which can be synthesised from pyruvate, did not show significant increases in label incorporation from [U-$^{13}$C]glucose in co-culture compared to each cell type cultured alone for all the human AML cell lines, with the exception of alanine and lactate in HL-60. Additionally, an increase in labelled acetate, alanine and lactate could be observed intracellularly in MS-5 in co-culture compared to MS-5 alone (*Figure 3E*). However, only label incorporation in lactate was significant confirming that glycolysis is upregulated in MS-5 cells in co-culture Overall, these results are in agreement with the transcriptomic data (*Figure 3C*), highlighting that glucose metabolism is upregulated in co-culture but also confirming that acetate derives from glycolysis.

## AML cells rewire stromal cell metabolism transferring ROS to obtain acetate

Tracer-based data on MS-5 cells in co-culture revealed that acetate secreted in co-culture derives from glucose (*Figure 3D–E*). However, transcriptomic data on MS-5 cells in co-culture did not show any upregulation of pyruvate dehydrogenase (PDH), which could convert glucose-derived pyruvate into acetate via acetyl-CoA (*Figure 3C*). Moreover, acetate secretion was also observed in MS-5 cells grown in thiamine-free media (*Figure 4—figure supplement 1A*), confirming that the acetate secretion was not dependent on keto acid dehydrogenases (*Liu et al., 2018*). An alternative mechanism of acetate synthesis involving a non-enzymatic oxidative decarboxylation of pyruvate into acetate has previously been described (*Liu et al., 2018*; *Vysochan et al., 2017*; *Tiziani et al., 2009*). This mechanism was reported to be mediated by ROS in mammalian cells and was linked to cells prone to overflow metabolism under the influence of high rates of glycolysis and excess pyruvate. Hence, we investigated whether ROS might play a role in acetate secretion in our co-culture system.

We first modulated ROS levels in human AML cell lines and MS-5 cells cultured alone and in co-culture and measured acetate production. Hydrogen peroxide was used to increase ROS levels and N-acetylcysteine (NAC) was used as a ROS scavenger. Extracellular acetate levels were measured by $^1$H-NMR after 24 hr. Increasing ROS levels with peroxide resulted in a significant increase in acetate production, particularly in SKM-1 and Kasumi-1 cells in co-culture (*Figure 4A* and *Figure 4—figure supplement 1B*). This experiment could not be carried out with HL-60 cells as peroxide treatment severely impaired the viability of HL-60 cells, as previously reported (*Nogueira-Pedro et al., 2013*). Additionally, when the ROS scavenger NAC was used, a decrease in the levels of acetate in both MS-5 cells cultured alone and in all the co-cultured cell lines was observed (*Figure 4A* and *Figure 4—figure supplement 1B*). The decrease in acetate levels was additionally confirmed using the ROS-scavenging enzyme, catalase (*Figure 4—figure supplement 1C*), indicating that acetate synthesis in MS-5 cells in co-culture is mediated by ROS.

Next, we compared intracellular ROS levels in AML and MS-5 cells in co-culture and cultured alone by labelling cells with the $H_2$DCFDA fluorescent dye. Our data showed that ROS levels in the three human AML cell lines used were significantly decreased in co-culture, whilst in the stromal cells ROS levels were significantly increased in two of the three co-cultures (*Figure 4B*). These results

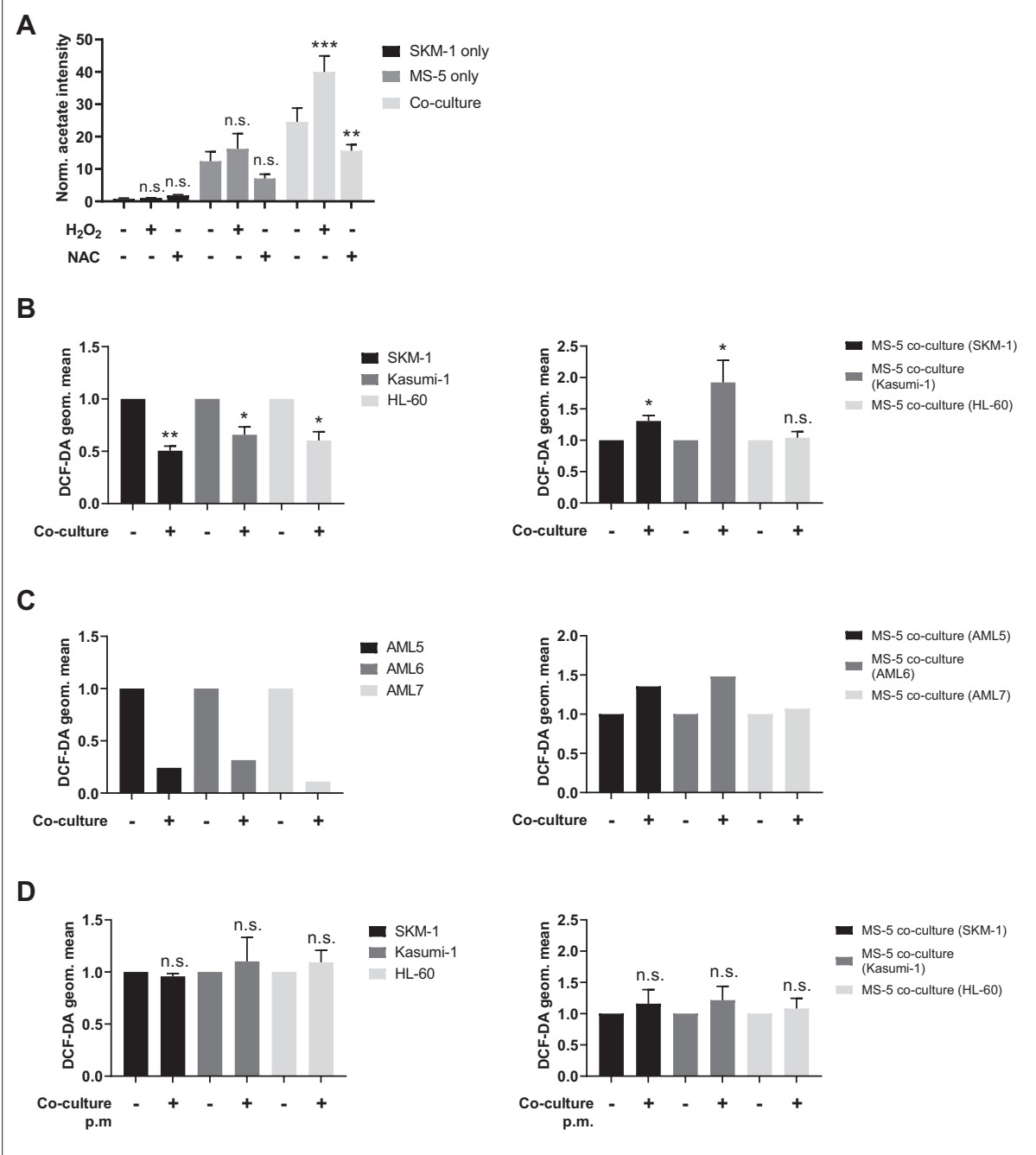

**Figure 4.** Acetate secretion is linked to ROS transfer from AML to stromal cells. (**A**) Extracellular acetate levels in SKM-1 (black) and MS-5 cells cultured alone (dark grey) and in co-culture (light grey) for 24 hr in a control medium, medium with 50 µM $H_2O_2$ or medium with 5 mM NAC. (**B**, **C** and **D**) Intracellular ROS levels measured by $H_2DCFDA$ staining in **B** and **D** AML cells or **C** primary patient-derived AML cells and MS-5 cells cultured alone and in co-culture in **B** and **C** direct contact. For **A**, and **B**, bars represent the mean of n=3 independent experiments and error bars represent standard deviations. For **A**, a Dunnett's multiple comparisons test was performed comparing each condition ($H_2O_2$/NAC) was compared to untreated; for **B** and **D**, an unpaired t test with Welch's correction was applied comparing co-culture conditions to cells cultured alone. p-Values are represented by n.s. for not significant, * for p-value <0.05, and ** for p-value <0.01.

The online version of this article includes the following source data and figure supplement(s) for figure 4:

**Source data 1.** Values and tats for panels included in *Figure 4*.

**Figure supplement 1.** Acetate secretion in thiamine free medium and after modulating ROS levels.

**Figure supplement 1—source data 1.** Acetate values and stats for MS5 cells in thiamine-free medium vs control (A), for Kasumi and HL-60 +/-NAC or H2O2 alone vs coculture (B), and for MS-5 cells with different concentrations of catalase (C).

suggested that AML cells might transfer ROS to stromal cells. We also performed the same experiment using three primary AML samples to corroborate the previous result. Fluorescence analysis showed decreased ROS levels in AML samples in co-culture and increased ROS levels in MS-5 cells in co-culture for the three primary AML samples analysed (*Figure 4C*), suggesting ROS transfer from AML cells to stromal cells in co-culture.

To further test whether AML cells might transfer ROS through a contact-dependent mechanism, we compared the intracellular ROS levels in AML and stromal cells cultured alone and co-cultured without direct contact using a permeable membrane. Fluorescence measurements in both cell types revealed that ROS levels remained unaltered in contact-free co-cultures (*Figure 4D*), indicating that ROS transfer could only occur via a contact-dependent mechanism.

Our results indicate that AML cells transfer ROS to stromal cells leading to acetate production, and that the ROS transfer and acetate production is dependent on contact between the two cell types.

## AML cells rewire stromal cell metabolism transferring ROS via gap junctions to obtain acetate

It has previously reported that haematopoietic stem cells can transfer ROS to stromal cells via gap junction to prevent senescence (*Taniguchi Ishikawa et al., 2012*), thus, we decided to examine gap junction genes in the transcriptome of MS-5 cultured alone and in co-culture. When individually examining the gap junction genes in MS-5 cells cultured alone vs co-culture, we found several gap junction genes upregulated in co-culture such as *Gja5, Gja8, Gjb5,* and *Gjc2* (*Figure 5A*). These results suggest that AML cells might establish gap junction interactions to transfer ROS to MS-5 cells when in co-culture. To test this hypothesis, we used the calcein-AM dye, which can only be transferred via gap junctions (*Kouzi et al., 2020*). We labelled MS-5 cells with calcein-AM, cultured them with unlabelled AML cells and analysed the fluorescence of AML cells after three hours. We found that, for the three human AML cell lines tested, the percentage of cells that had incorporated the calcein-AM dye from MS-5 cells was larger than 80% (*Figure 5B* and *Figure 5—figure supplement 1A*), indicating that AML cells can establish gap junctions with stromal cells when co-cultured in direct contact.

Next, we decided to confirm that AML cells can transfer ROS via gap junctions by inhibiting the gap junctions using carbenoxolone (CBX), a well-known gap junction inhibitor (*Kouzi et al., 2020*; *Davidson et al., 1986*; *Davidson and Baumgarten, 1988*). We first confirmed that efficiency of inhibition by analysing the calcein-AM dye transfer in a control and treated co-culture of the three human AML cell lines and MS-5 cells. The CBX treatment significantly reduced fluorescence levels and the percentage of cells with calcein-AM for all the human AML cell lines (*Figure 5C* and *Figure 5—figure supplement 1B*). We then compared intracellular ROS levels in both AML and stromal cells treated with CBX. CBX treatment abrogated the decrease in ROS levels in the three human AML cell lines (*Figure 5D*) and the increase of ROS levels in MS-5 mouse stromal cells (*Figure 5E*), indicating inhibition of ROS transfer in CBX treated co-cultures.

To get definitive proof that the formation of gap junctions is required for the metabolic re-wiring of stromal cells, we measured acetate production in AML cells and MS-5 cells cultured alone and in co-culture in the presence of CBX. Extracellular acetate levels were measured by $^1$H-NMR after 24 hr. We found that inhibition of gap junctions by CBX treatment resulted in a decrease in acetate levels when mouse stromal cells were co-cultured with human AML cell lines compared to untreated co-cultures, indicating that indeed acetate secretion is dependent on the formation of gap junctions between AML and stromal cells (*Figure 5F*).

Moreover, to determine whether in an in vivo setting, the inhibition of gap junctions could counteract the effect of ROS in leukaemia progression, we turned to our MLL-AF9 mouse model. C57BL/6 J WT mice were transplanted with BM of MLL-AF9[+] mice and then treated either with a potent ROS inducer, tert-Butyl hydroperoxide (TBHP) (*Kumar, 2007*; *Fatemi et al., 2013*), or with TBHP in combination with CBX. Our results showed that TBHP treatment accelerated the development of overt AML, reducing the survival of the recipient mice, with the number of monocytes in blood five times higher than vehicle-treated leukaemic mice at the end-point analysis (*Figure 5G and H*). In contrast, mice receiving both, TBHP and CBX, displayed survival and monocyte counts similar to vehicle-injected leukaemic mice. These experiments indicate that gap junction inhibition at least partially reduces the enhancing effect of ROS in the development of AML.

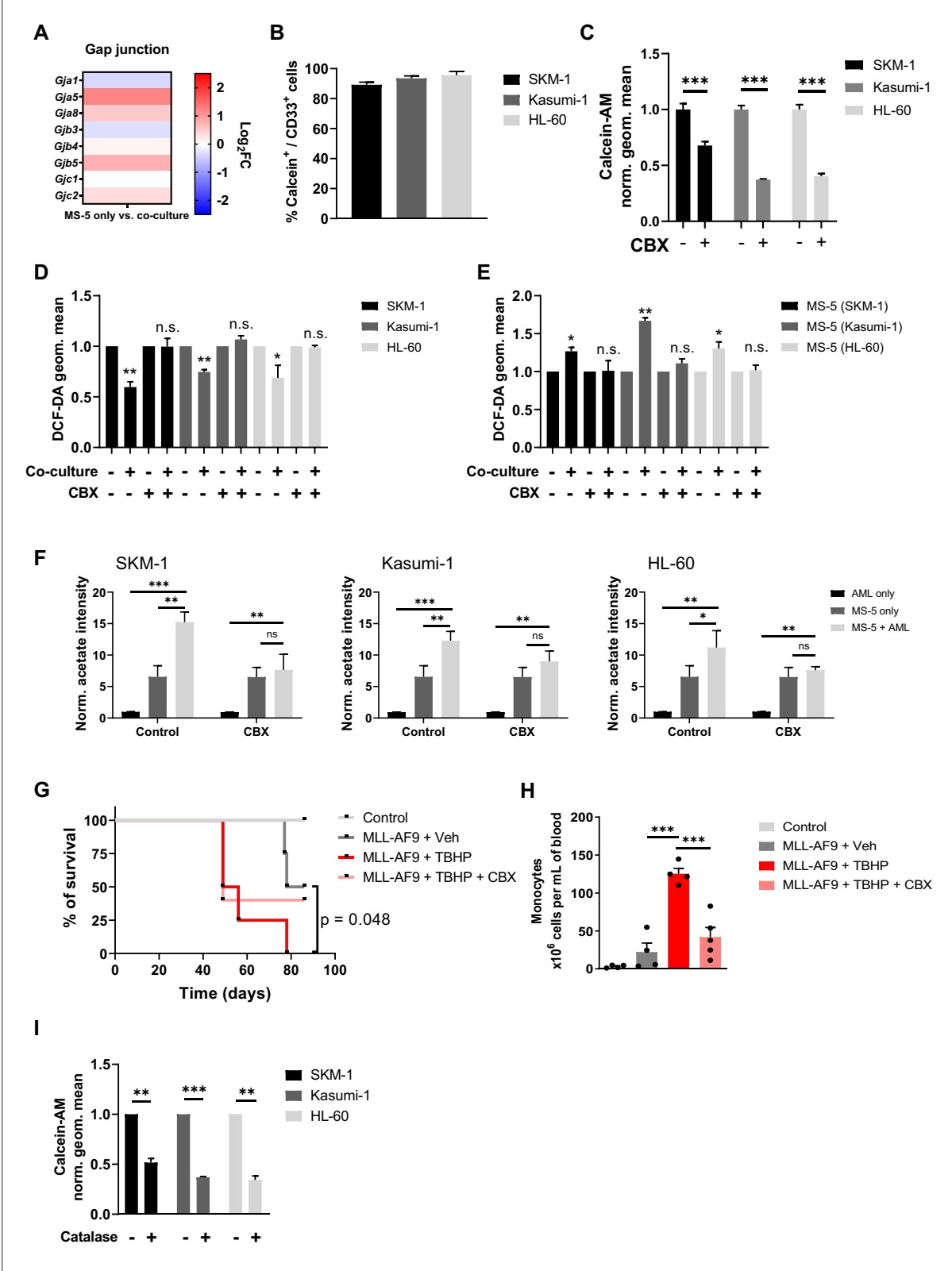

**Figure 5.** Acetate secretion is linked to ROS transfer from AML to stromal cells via gap junctions. (**A**) Fold change (FC) values of detected gene transcripts (TPMs) related to gap junctions. FC values are represented as $log_2FC$, red values indicate upregulation and blue values indicate downregulation in MS-5 cells in co-culture. (**B**) Frequencies of Calcein-AM and CD33 positive AML cell lines after being in co-culture with Calcein-AM stained MS-5 cells for 3 hr . (**C**) Geometric mean of Calcein-AM fluorescence in CD33 positive AML cells (treated or untreated with 200 μM

*Figure 5 continued on next page*

Figure 5 continued

carbenoxolone for 24 hr) after being in co-culture with Calcein-AM stained MS-5 cells for 3 hr . (**D** and **E**) Intracellular ROS levels measured by $H_2DCFDA$ staining in **D** AML cells and **E** MS-5 cells cultured alone and in co-culture in direct contact treated or untreated with 200 μM carbenoxolone for 24 hr. (**F**) Extracellular acetate levels in AML cells (black) and MS-5 cells cultured alone (dark grey) and in co-culture (light grey) for 24 hr in a control medium, or medium with 200 μM CBX. (**G**) Survival rate of C57BL6/J mice transplanted with bone marrow nucleated cells isolated from WT control or MLL-AF9[+] leukemic donors, treated with 500 μmol/kg of ROS-inducer tert-Butyl hydroxyperoxide (TBHP) alone or in combination with 30 mg/kg of gap-junction inhibitor carbenoxolone (CBX). Statistics indicate results of log-rank test for comparisons of Kaplan-Meier survival curves versus recipients of MLL-AF9[+] cells treated with vehicle. (**H**) Number of monocytes per mL of peripheral blood (PB) at endpoint analysis, measured with Procyte hematological analyzer (IDEXX BioAnalytics). (**I**), Geometric mean of Calcein-AM fluorescence in CD33 positive AML cells (treated or untreated with 500 μM catalase for 24 hr) after being in co-culture with Calcein-AM stained MS-5 cells for 3 hr. For (B, **C**, **D**, **E**, **F**, **H** and **I**) bars represent the mean of n=3 independent experiments and error bars represent standard deviations. For (**C**, **D**, **E**, **F**, **H** and **I**) an unpaired Student's t-test was applied for each condition. For **G**, a Gehan–Breslow–Wilcoxon test was applied. p-Values are represented by n.s. for not significant, * for p-value <0.05, ** for p-value <0.01 and *** for p-value <0.001.

The online version of this article includes the following source data and figure supplement(s) for figure 5:

**Source data 1.** Values and stats for all panels included in *Figure 5*.

**Figure supplement 1.** Calcein-AM and CD33 staining in SKM-1.

**Figure supplement 1—source data 1.** Percentage and stats for Calcein-AM +CD33 cells.

Once confirmed that acetate secretion is dependent of direct contact through gap junctions and ROS, we then sought to get further insight in the order of events and determine whether gap junctions are upregulated first, leading to increase ROS intake and metabolic re-wiring or whether the acetate production and gap junction formation were due to increased ROS. To discern between these two possibilities, we labelled MS-5 mouse stromal cells with calcein-AM, cultured them with unlabelled human AML cells in the presence of catalase and analysed the fluorescence of AML cells after three hours. We found that, for the three human AML cell lines tested, the calcein-AM transfer was reduced between 50 and 70% compared to cells co-cultured without the ROS scavenger (*Figure 5I*), indicating that ROS is required for the establishment of gap junctions between AML and stromal cells.

Overall, our results indicate that ROS is required for the upregulation of gap junctions facilitating in this way its transfer to stromal cells for acetate production and that inhibition of gap junctions affects acetate secretion by stromal cells (*Figure 6*).

## Discussion

AML cells are known to interact and remodel niche cells through various mechanisms, including the secretion of soluble factors, cytokines or metabolites, resulting in a better support of AML cells at the expense of normal haematopoiesis. Yet, metabolic crosstalk studies between AML and stromal cells remain scarce. Here, we have identified a novel metabolic communication between AML and stromal cells mediated by acetate. Our in vivo data shows high acetate levels in the bone marrow extracellular fluid of AML but not in healthy control mice, indicative of a potential role for acetate in AML development. Our in vitro data suggests that AML cells can modulate stromal cells into secreting acetate in co-culture, by rewiring stromal cell metabolism, and can then utilise acetate to feed their own TCA cycle and for lipid biosynthesis. Mechanistically, our data revealed that acetate secretion involves not only a higher glycolytic rate in stromal cells but is also a likely consequence of the non enzymatic ROS-mediated conversion of pyruvate to acetate., which was supported by the fact that MS-5 cells grown in thiamine-free media were still capable to produce acetate. Furthermore, our data indicates that AML cells can diminish their ROS levels by establishing gap junctions with stromal cells facilitating ROS transfer to stromal cells.

Studying the interactions between cancer cells and their microenvironment in terms of metabolism has become an exciting new field of cancer research. Our data indicates that AML cells can influence the metabolism of stromal cells causing increased acetate secretion, which was not observed in healthy counterparts.

We present several lines of evidence suggesting that stromal cells but not AML cells are responsible for acetate secretion: (i) stromal cells cultured alone secreted acetate, whilst AML cell lines and primary AML cells did not; (ii) after separating stromal cells from co-culture with AML cell lines, stromal cells continued to secrete acetate at a similar rate to that observed in co-culture; and, (iii) glycolysis was

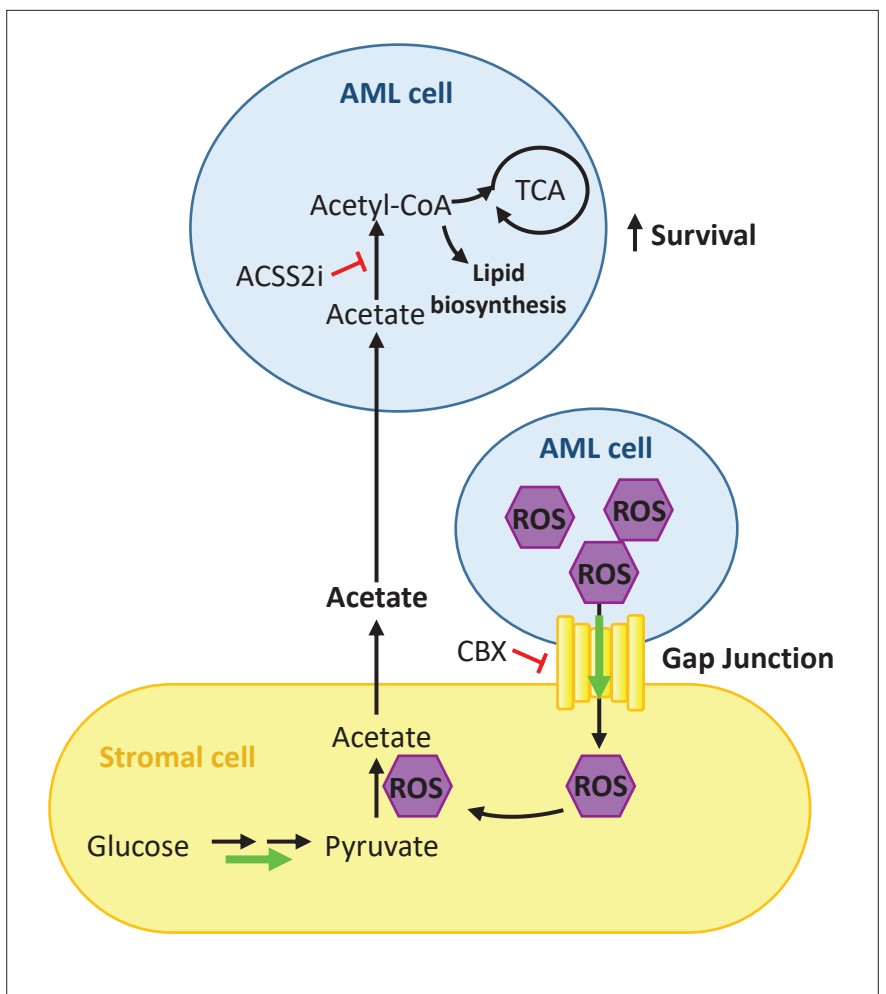

**Figure 6.** Schematic summary of our findings. AML cells present high levels of ROS that activate the formation of gap junctions. ROS is transferred from AML cells to stromal cells through these gap junctions and it is utilised by stromal cells to convert pyruvate into acetate. Acetate is secreted from the stromal cells and uptake it by AML cells. In AML cells, imported acetate is used to feed the TCA cycle as well as for lipid biosynthesis, promoting the survival of the AML cell.

upregulated in stromal cells in co-culture, and glucose was found to be the precursor of the secreted acetate in co-culture. Thus, although some alternative metabolic rewiring in AML cells dependent on the direct contact with stromal cells cannot be discarded, our data strongly suggests stromal cells secreting acetate as the main pathway. Although we cannot provide an absolute proof that AML cells consume the acetate specifically secreted by stromal cells, our data adding [13]C-labelled acetate to co-cultured AML cells and showing label incorporation in several TCA cycle metabolites and in lipid metabolism exclusively in AML cells, strongly supports this hypothesis. Interestingly, acetate has been reported as an alternative fuel for cancer cells (*Lyssiotis and Cantley, 2014*; *Comerford et al., 2014*), especially under low oxygen conditions or lipid depletion (*Gao et al., 2016*; *Schug et al., 2015*; *Yoshii et al., 2009*), but it has not previously been described to participate in crosstalk between any type of cancer cells and their microenvironment. Nonetheless, other monocarboxylate metabolites, such as lactate and alanine, have been reported to participate in different types of metabolic interplay between stromal and cancer cells (*Sousa et al., 2016*; *Sonveaux et al., 2008*; *Whitaker-Menezes et al., 2011*). Our work reveals that aside from lactate and alanine, other monocarboxylate metabolites, such as acetate, can be utilised by leukaemic cells as a biofuel. Although acetate usage to feed the TCA cycle had already been described in AML (*Saborano et al., 2019*) and other types of cancer (*Schug et al., 2015*; *Mashimo et al., 2014*), this is, to our knowledge, the first report of AML cells utilising acetate secreted by stromal cells in co-culture.

We have also proposed a mechanism for the altered stromal cellular metabolism, involving increased glycolysis and the ROS-mediated chemical conversion of pyruvate to acetate. Transcriptomic data showed that glycolysis is upregulated in stromal cells in co-culture, and tracer-based metabolism using [U-$^{13}$C]glucose, demonstrated that more label gets incorporated into acetate in stromal cells in co-culture compared to either AML or stromal cells cultured alone. Enrichment of hypoxia genes and elevated Pdk expression have been reported to be related to higher glycolytic activity (*Testa et al., 2016*; *Kocabas et al., 2015*; *Takubo et al., 2013*). Additionally, published single-cell transcriptomic data of stroma cells in vivo by the group of Scadden, has shown three specific clusters which have upregulated glycolysis and hypoxia pathways (*Baryawno et al., 2019*). In particular, the osteoprogenitor cluster was shown to be upregulated in leukaemic mice and shared the same hallmark gene sets that MS-5 in co-culture (TNFα, Hypoxia, allograft rejection, glycolysis, IFNγ, K-RAS, complement, epithelial to mesenchymal transition and IL2-STAT5) (*Baryawno et al., 2019*). Similarly, other cancer cells *Shan et al., 2017*; *Migneco et al., 2010*; *Pavlides et al., 2009*; *Cruz-Bermúdez et al., 2019* have also been shown to modulate stromal metabolism by increasing glycolysis. The proposed mechanism, involving higher glycolysis and pyruvate conversion to acetate via ROS, was further supported by increasing and lowering ROS levels in co-culture in our studies. Lowering ROS levels reduced the acetate secretion of MS-5 cells in co-culture to similar levels as when MS-5 cells are cultured alone, highlighting the implication of ROS in acetate secretion in co-culture.

AML cells are known to exhibit high levels of ROS (*Hole et al., 2013*; *Robinson et al., 2020*). However, our data has shown that AML cells in co-culture present lower levels of ROS than AML cells cultured alone, which is in agreement with recent studies in primary AML and mesenchymal stromal cell co-cultures (*Forte et al., 2020*; *Vignon et al., 2020*). The current mechanisms to describe this phenomenon involve mitochondrial transfer and activation of glutathione-related antioxidant pathways (*Forte et al., 2020*; *Vignon et al., 2020*), although previous data on haematopoietic stem cells (HSCs) revealed that HSCs can directly transfer ROS via gap junctions to stromal cells (*Taniguchi Ishikawa et al., 2012*). Interestingly, our data showed that the decrease in ROS levels was counteracted by inhibiting contact between MS-5 and AML cells using a permeable membrane. We also found several gap junction genes upregulated in stromal cells in co-culture. Moreover, we prevented ROS transfer from AML to stromal cells by using the gap junction inhibitor CBX (*Kouzi et al., 2020*; *Davidson et al., 1986*; *Davidson and Baumgarten, 1988*), which abolished the increase in extracellular acetate level in co-cultures in vitro and partially reduced the effect of ROS inducers (TBHP) in the aggravation of AML in vivo. Thus, our results suggest that ROS transfer via gap junctions, at least partially, mediates the mechanism behind AML cells presenting lower levels of ROS in co-culture.

Overall, this work reveals a unique and novel metabolic communication between AML and stromal cells that involves acetate as the main crosstalk metabolite. We showed that AML cells are capable of modulating the metabolism of stromal cells by transferring ROS via gap junctions resulting in an increased secretion of acetate and its subsequent accumulation in the extracellular medium, which correlates with the observation of higher acetate levels in the bone marrow extracellular fluid of AML mice compared to control mice. Furthermore, we found that AML cells can consume acetate and use it to feed the TCA cycle and for lipid biosynthesis. We believe our findings provide a better understanding of how AML cells communicate with stromal cells and could serve as a basis for the development of novel therapeutic strategies to target AML cells by modulating gap junction formation or modulating acetate use as an adjuvant therapy.

## Materials and methods

**Key resources table**

| Reagent type (species) or resource | Designation | Source or reference | Identifiers | Additional information |
|---|---|---|---|---|
| Strain, strain background (*Mus musculus*, females) | C57BL/6 J | The Jackson Laboratory | RRID:IMSR_JAX:000664 | |
| Strain, strain background (*Mus musculus*, males) | MLL-AF9 | The Jackson Laboratory | RRID:IMSR_JAX:009079 | Wild type littermates used as controls |

*Continued on next page*

*Continued*

| Reagent type (species) or resource | Designation | Source or reference | Identifiers | Additional information |
|---|---|---|---|---|
| Cell line (*Homo sapiens*) | SKM-1 | DMSZ | DMSZ ACC 547; RRID: CVCL_0098 | |
| Cell line (*Homo sapiens*) | Kasumi-1 | DMSZ and a gift from the laboratory of C.Bonifer (UoB) | DMSZ ACC 220; RRID: CVCL_0589 | |
| Cell line (*Homo sapiens*) | HL-60 | ATCC | ATCC CCL-240 RRID: CVCL_0002 | |
| Cell line (*Mus musculus*) | MS-5 | Gift from the laboratory of JJ Schuringa (Groninger University) | DMSZ ACC 441 RRID:CVCL_2128 | |
| Cell line (*Homo sapiens*) | HeLa | ATCC | ATCC CCL-2 RRID: CVCL_0030 | |
| Antibody | CD33 (mouse monoclonal, clone P67.6) | eBioscience | 48-0337-42 | 1:100 |
| Antibody | anti-CD34 APC (mouse monoclonal) | BD Biosciences | 560940 | 10 ul per 10 million cells |
| Antibody | anti-CD38 FITC (mouse monoclonal) | BD Biosciences | 555459 | 10 ul per 10 million cells |
| Chemical compound, drug | tert-Butyl hydroxyperoxide (TBHP) | Sigma/Merck | 478139 | |
| Chemical compound, drug | carbenoxolone (CBX) | Sigma-Aldrich | C4790 | For in vitro experiments |
| Chemical compound, drug | carbenoxolone (CBX) | Thermofisher | J63714.03 | For in vivo experiments |
| Chemical compound, drug | Catalase | Sigma-Aldrich | C100 | |
| Chemical compound, drug | 2',7'-Dichlorofluorescin diacetate (DCFH-DA) | Sigma-Aldrich | D6883 | |
| Chemical compound, drug | $H_2O_2$ | Merck | 386790 | |
| Chemical compound, drug | N-acetylcysteine (NAC) | Merck | 106425 | |
| Chemical compound, drug | Calcein-AM green | Invitrogen | C1430 | |
| Chemical compound, drug | CellTrace carboxyfluorescein succinimidyl ester (CFSE) | Invitrogen | C34570 | |
| Chemical compound, drug | ACSS2i | Selleck | S8588 | |
| Chemical compound, drug | Cytarabine (AraC) | Sigma-Aldrich | C1768 | |
| Chemical compound, drug | [U-$^{13}$C]Glucose | CortecNet | CC860P1 | |
| Chemical compound, drug | [2-$^{13}$C]acetate | Sigma-Aldrich | 279315–1 G | |
| Other | CD34 magnetic microbeads | Miltenyi Biotec | 130-046-702 | Please see Materials and Methods section under Primary Patient samples |
| Other | CD117 magnetic microbeads | Miltenyi Biotec | 130-091-332 | Please see Materials and Methods section under Primary Patient samples |
| Sequence-based reagent | *Hk2* q PCR primers | Sigma-Aldrich | NM_013820 | KiCqStart primers KSPQ12012 |
| Sequence-based reagent | *Pdhx* q PCR primers | Sigma-Aldrich | NM_175094 | KiCqStart primers KSPQ12012 |
| Sequence-based reagent | *Pdk1* q PCR primers | Sigma-Aldrich | NM_172665 | KiCqStart primers KSPQ12012 |

*Continued on next page*

*Continued*

| Reagent type (species) or resource | Designation | Source or reference | Identifiers | Additional information |
|---|---|---|---|---|
| Sequence-based reagent | *Pdk2* q PCR primers | Sigma-Aldrich | NM_133667 | KiCqStart primers KSPQ12012 |
| Sequence-based reagent | *B2m* qPCR primers | Thermo Fisher Scientific | NM_009735.3 | |
| Software.algorithm | Flowjo | BD-Bioscience | | |
| Software.algorithm | MetaboLab software within the MATLAB environment (MathWorks). | *Ludwig and Günther, 2011*<br><br>. https://doi.org/10.1186/1471-2105-12-366 | | https://www.ludwiglab.org/software-development |
| Software.algorithm | Chenomx 7.0 software (Chenomx Inc) | Chenomx Inc https://www.chenomx.com/ | | |
| Software.algorithm | FastQC 0.11.7 software | http://www.bioinformatics.babraham.ac.uk/projects/fastqc/ | | |
| Software.algorithm | Kallisto 0.43.0 software | *Bray et al., 2016* | | https://pachterlab.github.io/kallisto/ |
| Software.algorithm | R statistical package Sleuth 0.30.0 | *Pimentel et al., 2017* | | https://github.com/pachterlab/sleuth |
| Software.algorithm | the R statistical package fgsea 1.10.0 | http://bioconductor.org/packages/release/bioc/html/fgsea.html | | |
| Software.algorithm | R statistical package BioMart 2.40.3 | https://bioconductor.org/packages/biomaRt | | |
| Software.algorithm | GraphPad version 8 | https://www.graphpad.com/ | | |

## Cell lines

The human AML cell lines (SKM-1, Kasumi-1, and HL-60), the mouse stromal cell line (MS-5) and the human cervical cancer cell line (HeLa) were cultured in RPMI 1640 media supplemented with 15% (v/v) FBS, 2 mmol/L L-glutamine and 100 U/ml Penicillin/Streptomycin (all from Thermo Fisher Scientific). Co-cultures were plated in a 4:1 AML-stromal ratio, and 750,000 cells/ml density of AML cells over confluent stromal cells. Prior to cell extraction for NMR, RNA collection or protein extraction, a suspension of 10 million AML cells was collected, the stromal layer was washed with PBS and was subject to mild trypsinisation with 1:5 dilution of 0.25% Trypsin 1 mM EDTA (Thermofisher) to remove attached residual leukaemic cells before completely detaching stromal cells with 0.25% Trypsin-EDTA. Micoplasma test using MycoAlert (Lonza) was performed every three months in all cell lines.

## Primary patient samples

AML and PBMC primary specimens' procedures were obtained in accordance with the Declaration of Helsinki at the University Medical Center Groningen, approved by the UMCG Medical Ethical Committee or at the University Hospital Birmingham NHS Foundation Trust, approved by the West Midlands – Solihull Research Ethics Committee (10 /H1206//58).

Additional information about the primary AML samples used in this study can be found in *Supplementary file 1*. Peripheral blood (PB) and bone marrow samples from AML patients and healthy donors were obtained in heparin-coated vacutainers. Mononuclear cells were isolated using Ficoll-Paque (GE Healthcare) and stored at –80 °C.

For AML1-2 and PBMC1-2, samples were thawed and resuspended in newborn calf serum (NCS) supplemented with DNase I (20 Units/mL) (Roche), 4 mM $MgSO_4$ (Sigma-Aldrich) and heparin (5 Units/mL) (Ziekenhuis Apotheek Midden-Brabant) and incubated on 37 °C for 15 min. For AML1 and PBMC1, cells were sorted after thawing by fluorescence-activated cell sorting (FACS) for the CD34 +CD38- population using 10 µL of anti-CD34 APC (560940, BD Biosciences), 10 µL of anti-CD38 FITC (555459, BD Biosciences) and 10 µL of of DAPI (D1306, Thermo Fisher Scientific) per 10 million

cells. Cells were sorted using a Sony SH800S (Sony) sorter. For AML2 and PBMC2, cells were thawed and the CD34 +population was sorted by magnetic-activated cell sorting (MACS) using 10 μL of anti-CD34 microbeads (130-046-702, Miltenyi) per million of expected CD34 cells following manufacturer's protocol.

AML1-2, PBMC1-2, and MS-5 cells were cultured alone and in co-culture in a 4:1 AML/PBMC-stromal ratio and 500,000 cells/mL density in α-MEM (Gibco) with 12.5% (v/v) FCS (Sigma-Aldrich), 1% (v/v) Pen/Strep (Thermo Fisher Scientific), 12.5% (v/v) Horse serum (Invitrogen), 0.4% (v/v) β-mer-captoethanol (Merck Sharp & Dohme BV) and 0.1% hydrocortisone (H0888, Sigma-Aldrich). For AML1 and AML2, the medium was supplemented with 0.02 μg/mL of IL-3 (Sandoz), 0.02 μg/mL NPlate (Amgen), and 0.02 μg/mL of G-CSF (Amgen). CD34$^+$ cells from PBMC1 and PBMC2 the medium was supplemented with 100 ng/ml of human SCF (255-SC, Novus Biologicals), 100 ng/ml of NPlate , 100 ng/ml of FLT3 ligand (Amgen) and 20 ng/ml of IL-3. Samples of medium were collected at 0 and 48 hr.

AML3 - 7 and PBMC3 were thawed and kept in culture for 16–24 hr in Stem Span H3000 media (STEMCELL Technologies) supplemented with 50 μg/ml ascorbic acid (Sigma-Aldrich), 50 ng/ml human SCF (255-SC-010, R&D Systems), 10 ng/ml human IL-3 (203-IL-010, R&D Systems), 2 units/ml human-erythropoietin (100–64, PeproTech), 40 ng/ml insulin-like growth factor 1 (IGF-1) (100–12, PeproTech), 1 μM dexamethasone (D2915, Sigma-Aldrich), and 100 μg/ml primocin (ant-pm-2, Invivogen). CD34 +cKit + cells from AML and healthy controls were purified using magnetic microbeads (130-046-702 (CD34) and 130-091-332 (CD117), Miltenyi Biotec). Cells were cultured in the supplemented Stem Span media for 24 hr prior to co-culture setting. Co-cultures were plated with a 4:1 leukaemic to stromal cells ratio and a 300,000 cells/ml density in supplemented Stem Span medium. Samples of leukaemic/healthy cells and MS-5 cells alone were also cultured in supplemented Stem Span media as controls. Samples of medium were collected at 0 and 48 hr.

## In vivo experiments
### Animals and transplantation
Twelve weeks old C57BL6/J female mice (The Jackson Laboratory) were lethally irradiated with 9 Gy (2 doses of 4.5 Gy separated by 3 hr) using an X-RAY source (Rad Source's RS 2000). Mice were transplanted by intravenous injection 4 hr after with 2x10$^6$ bone marrow (BM) nucleated cells isolated as previously described (*Arranz et al., 2014*) from leukemic male mice heterozygous for MLL-AF9 knock-in fusion transgene or wild-type (WT) control male littermates (*Corral et al., 1996*). Male transgenic and WT control MLL-AF9 were purchased from The Jackson Laboratory (Stock No: 009079) and were 6 months old when euthanized to allow development of signs of leukemic transformation driven by the MLL-AF9 fusion oncogene in transgenic animals. MLL-AF9 expression in BM cells derived from transgenic donors results in development of AML with high blast counts in recipients.

### Extraction of bone marrow extracellular fluid
Recipients were euthanized 6 months after the transplant, bone marrow from femur and tibia was flushed with a syringe in 150 μL of cold PBS and centrifuged at 15,000 g for 10 min at 4 °C. The supernatant made of bone marrow extracellular fluid (BMEF) was kept for analysis of acetate level.

## Pharmacological treatments
Treatments started 6 weeks after transplantation of MLL-AF9$^+$ or WT BM nucleated cells into C57BL6/J female recipients. Mice were daily injected intraperitoneally with 500 μmol/kg of ROS enhancer tert-Butyl hydroxyperoxide (TBHP) or PBS as described previously (*Fatemi et al., 2013*). A group of mice were treated with TBHP and in addition injected intraperitoneally every other day (Monday, Wednesday and Friday) with 30 mg/kg of gap-juction inhibitor carbenoxolone (CBX) (*Kouzi et al., 2020*). Animals were terminated when they reached 100x10$^6$ white blood cells per mL of PB or when moribund. Eight days after a whole experimental group was terminated, the rest of animals were terminated. Time elapsed between the start of the treatment and termination was used for Kaplan-Meier analysis of survival rate. Peripheral blood was obtained from the heart at the endpoint and analyzed using a Procyte hematological analyzer (IDEXX BioAnalytics).

Sample size was calculated according to standard deviations from the means of parameters in groups under study, 5% significance level, power of 90%, and two-tailed T test. Standardised effect

size (signal/noise ratio) = (Mean1-Mean2)/SD is expected to be higher than 2.2. Randomization: Mice were randomized to treatment groups. Mice of the same sex and age were used to control for covariates. No blinding was performed due to regulations at the Animal Facilities of the UiT – The Arctic University of Norway and the University of Oslo. Animals that showed symptoms of unrelated disease were excluded of the study (1 mouse). Criteria applied for mouse termination before the established end point were in accordance with the Norwegian Food and Safety Authority. Outliers were not removed. ARRIVE guidelines were followed.

Experiments were conducted with the ethical approval of the Norwegian Food and Safety Authority. Animals were housed under specific opportunistic and pathogen free environment at the Section of Comparative Medicine at the University of Oslo, Norway, and the Unit of Comparative Medicine at UiT - the Arctic University of Norway.

## Proliferation analysis using CFSE

The CellTrace carboxyfluorescein succinimidyl ester (CFSE) Cell Proliferation Kit (C34570, Invitrogen) was used to assess proliferation of AML cells following the manufacturer's protocol. AML cells were stained and their fluorescence was assessed before dividing the bulk of cells into culturing them alone or with MS-5 cells for 48 hr. Small aliquots of cells after 24 and 48 hr were analysed by flow cytometry. Flow cytometry analysis was carried out in a CyAn ADP flow cytometer (Beckman Coulter). Data analysis was performed using the FlowJo software package (BD).

## Cellular ROS measurements using DCFH-DA

Cellular ROS was measured by incubating cells with 100 µM 2′,7′-Dichlorofluorescin diacetate (DCFH-DA) (D6883, Sigma-Aldrich) in Hank's Balanced Salt Solution (HBSS) (Thermo Fisher Scientific) at 37 °C for 30 min protected from light. Cells were then harvested and stained with 5 µg/µL anti-human CD33 eFluor 450 (eBioscience, P67.6) for 30 min at 4 °C before flow cytometry analysis as previously described.

## Thiamine-free medium comparison

Thiamine-free medium (R9011-01, United States Biological) was prepared as per manufacturer instructions and supplemented with 10% dialised FBS.

MS-5 cells were cultured in thiamine-free medium or control RPMI medium for 4 days prior to the experiment. Cells were then seeded with fresh thiamine-free or control RPMI medium and incubated for 24 hr. Samples of medium were collected at 0 and 24 hr and kept at –80 °C.

## ROS-related treatments with $H_2O_2$ and NAC

SKM-1, Kasumi-1, HL-60 and MS-5 cells cultured alone and in co-culture were incubated for 24 hr in 50 µM $H_2O_2$ (Merck) complete cell culture medium, 5 mM N-acetylcysteine (NAC) (106425, Merck) complete cell culture medium or control medium. Samples of medium were collected at 0 and 24 hr and kept at –80 °C.

## Calcein-AM dye transfer assay

Functional gap junction presence was evaluated using the fluorescent dye Calcein-AM green (Invitrogen, C1430) adapting a previously established protocol (*Kouzi et al., 2020*). MS-5 cells were stained with 500 nM Calcein-AM dye in complete cell culture medium for 1 hr at 37 °C. Stained cells were washed with serum-free medium for 30 min at 37 °C before being co-cultured with AML cells for 3 hr. AML cells were then harvested and stained with 5 µg/µL anti-human CD33 eFluor 450 (eBioscience, P67.6) for 30 min at 4 °C before flow cytometry analysis as previously described. Calcein-AM dye transfer was quantified as the frequency of $CD33^+$ and Calcein-$AM^+$ cells or as the geometric mean of FITC channel.

## Carbenoxolone and catalase treatments

Carbenoxolone (CBX) disodium salt (C4790, Sigma-Aldrich) was prepared fresh at 200 µM in cell culture medium. Catalase (C100, Sigma-aldrich) was prepared fresh at 10 mg/ml in PBS and filtered sterilized. Cells were resuspended in the CBX medium or media containing catalase at 100 µg/ml or

500 µg/ml and cultured alone or in co-culture for 24 hr prior to ROS, Calcein-AM dye transfer experiments or medium collection.

## Tracer-based NMR experiments

[U-$^{13}$C]Glucose (CC860P1, CortecNet) was added to RPMI 1640 medium without glucose (11879020, Merck) to a final concentration of 2 g/L (as in the complete cell culture medium) and was supplemented as usual with 15% (v/v) FBS, 2 mmol/L L-glutamine and 100 U/mL Pen/Strep. The medium was prepared fresh and was filtered with a 0.2 µm syringe filter (Sartorius) before each experiment.

4 mM sodium [2-$^{13}$C]acetate (279315–1 G, Sigma-Aldrich) was added to complete cell culture medium and the medium was filtered with a 0.2 µm syringe filter before each experiment. Cells were incubated for the time indicated in each experiment before separation of cells and/or metabolite extraction. Unlabelled samples were prepared as a control for 2D-NMR experiments and to measure acetate consumption by adding unlabelled sodium acetate trihydrate (1.37012, Merck) to complete cell culture medium before metabolite extraction or collection of medium samples.

## Metabolite extraction

Suspension cells and detached adherent cells were washed with PBS before being rapidly resuspended in 400 µl of HPLC grade methanol pre-chilled on dry ice. Samples were transferred to glass vials and were subject to 10:8:10 methanol-water-chloroform extraction as described in *Saborano et al., 2019*. Polar phase samples were evaporated in a SpeedVac concentrator; non-polar phase samples were evaporated over night in a fume hood. Samples were subsequently kept at –80 °C prior to sample preparation and analysis by NMR.

## Sample preparation for NMR

Medium samples or bone marrow extracellular fluid samples were mixed with a D$_2$O phosphate buffer containing TMSP ((3-trimethylsilyl)propionic-(2,2,3,3-d$_4$)-acid sodium salt) and NaN3 to obtain a final concentration of 0.1 M phosphate, 0.5 mM TMSP ((3-trimethylsilyl)propionic-(2,2,3,3-d$_4$)-acid sodium salt) and 1.5 mM NaN3 (all from Sigma). Samples were subsequently transferred to 3 mm NMR tubes. No sample derivation is required to detect acetate by NMR.

Dried polar extracts for tracer-based NMR experiments were reconstituted in 50 µL of 0.1 M phosphate buffer in 100% D$_2$O with 3 mM NaN$_3$ and 0.5 mM TMSP. Samples were sonicated for 10 min and transferred to 1.7 mm NMR tubes using a micro pipet system. Samples were prepared freshly before the acquisition.

Dried non-polar extracts for tracer-based NMR experiments were reconstituted in in CDCl$_3$ (1% TMSP) and transfer into 5 mm NMR tubes. Samples were prepared freshly before the acquisition.

## NMR acquisition and analysis

All NMR data were acquired on Bruker 600MHz spectrometers equipped with Avance-III consoles using cooled Bruker SampleJet autosamplers. For media samples, a 5 mm triple resonance cryoprobe (TCI) z-axis pulsed field gradient (PFG) cryogenic probe was used, and for cell extracts, a TCI 1.7 mm z-PFG cryogenic probe was used. Probes were equipped with a cooled SampleJet autosampler (Bruker) and automated tuning and matching.

For medium samples, spectra were acquired at 300 K using a 1D $^1$H-NOESY (Nuclear Overhauser effect spectroscopy) pulse sequence with pre-saturation water suppression (noesygppr1d, standard pulse sequence from Bruker). The spectral width was 12 ppm, the number of data points was TD 32,768, the interscan delay was 4 s. The $^1$H carrier was set on the water frequency and the $^1$H 90° pulse was calibrated at a power of 0.256 W and had a typical length of ca 7–8 µs. 64 scans and 8 dummy scans were acquired and the total experimental time was 7.5 min. Spectra were processed using the MetaboLab (*Ludwig and Günther, 2011*) software within the MATLAB environment (MathWorks). A 0.3 Hz exponential apodization function was applied to FIDs followed by zero-filling to 131,072 data points prior to Fourier transformation. The chemical shift was calibrated by referencing the TMSP signal to 0 ppm and spectra were manually phase corrected. The baseline was corrected by applying a spline baseline correction, the water and edge regions of the spectra were excluded before scaling the spectra using a probabilistic quotient normalization (PQN). Chenomx 7.0 software (Chenomx Inc) and the human metabolome database (HMDB) were used to assign the metabolites present in

the acquired spectra. Metabolite signal intensities were obtained directly from the spectra and were normalised to a control medium sample obtained at time 0 hr. For bone marrow extracellular fluid samples, metabolite signal areas were obtained directly from the spectra.

For $^{13}$C-filtered $^1$H-NMR experiments, [U-$^{13}$C]glucose labelled medium samples were analysed with $^{13}$C-filtered $^1$H-NMR spectroscopy as described in *Reed et al., 2019*. Spectra were acquired at 300 K using a double gradient BIRD filter pulse sequence developed in-house (*Reed et al., 2019*). A pulse program combining the $^1$H[$^{12}$C] and the all-$^1$H experiments in scan-interleaved mode was used. The difference between the two FIDs yields the $^1$H[$^{13}$C] signal. The spectral width was 12 ppm, the number of data points was 16,384, and the relaxation delay was 5.3 s. 256 scans with 64 dummy scans were acquired and the experimental time was 15 min. $^{13}$C-filtered $^1$H-NMR spectra were processed in Topspin 4.0.5 (Bruker). Spectra were zero filled to 32,768 data points before Fourier transformation. Phase correction was applied to the $^1$H[-$^{12}$C] and the all-$^1$H spectra and the difference $^1$H[$^{13}$C] spectrum was obtained by aligning on chemical shift. Metabolites were selected in the $^1$H[$^{13}$C] spectrum and integrated in all the spectra ($^1$H[$^{12}$C], $^1$H[$^{13}$C] and all-$^1$H). Label incorporations ($^{13}$C percentages) were calculated by dividing the signal areas obtained in the $^1$H[$^{13}$C] spectrum by the ones obtained in the all-$^1$H spectrum.

For $^1$H-$^{13}$C-HSQC experiments, spectra were acquired using a modified version of Bruker's hsqcg-phprsp pulse program, with additional gradient pulses during the INEPT echo periods and using soft 180° pulses for $^{13}$C. For the $^1$H dimension, the spectral width was 13.018 ppm with 2048 complex points as described by *Saborano et al., 2019*. For the $^{13}$C dimension, the spectral width was 160 ppm with 2048 complex points. Two scans were acquired per spectrum and with an interscan delay of 1.5 s. Non-uniform sampling (NUS) with a 25% sampling schedule (generated using the Wagner's schedule generator (Gerhard Wagner Lab, Harvard Medical School) with a tolerance of 0.01 and default values for the other parameters) was used with 4096 increments yielding 8192 complex points after processing. The total experimental time was 4 hr. Spectra were processed and phased with NMRPipe (National Institute of Standards and Technology of the U.S.; Appendix 8.1). MetaboLab was used to reference the chemical shift using the signal for the methyl group of L-lactic acid, at 1.31/22.9 ppm in the $^1$H and $^{13}$C dimensions, respectively. Identification of metabolites in 2D spectra was carried out using the MetaboLab software which includes a chemical shift library for ca. 200 metabolites. Intensities were obtained from signals in the spectra and were corrected for differences in cell numbers contributing to the sample as follows: A $^1$H-NMR spectrum was acquired for each sample and the total metabolite area of this spectrum after removal of the water and TMSP reference signal was calculated in MetaboLab. The intensities in the 2D spectra were then divided by the total metabolite area of the corresponding $^1$H-NMR spectrum. To obtain the % of $^{13}$C in a metabolite, the normalized intensity of a certain carbon in the labelled sample was divided by the normalized intensity of the same carbon in the unlabelled sample and was multiplied by the natural abundance of $^{13}$C (1.1%).

To determine the $^{13}$C-label incorporation in lipids, 1D $^1$H spectra (zgpr, standard pulse sequence from Bruker) were acquired on a Bruker 600MHz spectrometer equipped with a 5 mm triple resonance pulse field gradient (PFG) room temperature probe at 300 K. The data was collected with a spectral width of 12 ppm, the number of data points was 32,768 and interscan delay was 4 s. The $^1$H carrier was set at 7.26 ppm corresponding to CDCl$_3$. The spectra were processed and analyzed using Bruker Topspin. A 0.3 Hz exponential apodization function was applied to FIDs followed by zero-filling to 131,072 data points prior to Fourier transformation. The chemical shift was calibrated by referencing the TMSP signal to 0 ppm and spectra were manually phase corrected. The baseline was corrected by applying a spline baseline correction.

## RNA extraction and sequencing

MS-5 cells were cultured alone or in co-culture with SKM-1 cells for 24 hr. Cells were separated and washed with PBS prior to RNA extraction with TRIzol (Gibco) according to the manufacturer's protocol. RNA was purified with RNeasy Plus Micro kit. Samples were sent to Theragenetex to be sequenced with Novaseq 150 bp PE with 40 M reads.

## Real-time PCR

Samples of RNA from SKM-1 and MS-5 co-cultures were collected and extracted using Trizol (Invitrogen) following manufacturer's protocol. cDNA was synthesized using the M-MLV reverse transcriptase

(Promega) according to manufacturer's instructions. For gene expression analysis, qRT-PCR of *Hk2* (NM_013820), *Pdhx* (NM_175094), *Pdk1* (NM_172665), and *Pdk2* (NM_133667; all KiCqStart primers KSPQ12012, Sigma Aldrich) were carried out using the SYBRGreen Master mix (Thermo Fisher Scientific) and qRT-PCR of *B2m* (NM_009735.3, TaqMan assays, Thermo Fisher Scientific) was performed using TaqMan PCR Master Mix (Thermo Fisher Scientific). Reactions were performed in a Stratagene Mx3000P and were run in triplicate. Relative gene expression was calculated following the $2^{-\Delta\Delta Ct}$ method relative to the expression of *B2m*.

## RNA sequencing

RNA samples of MS-5 cells cultured alone and in co-culture with SKM-1 cells extracted using Trizol were purified using the Rneasy Plus Micro kit (Qiagen) according to manufacturer's protocol. Transcriptome analysis was performed by Theragen Etex Co., LTD. (https://www.theragenetex.net). cDNA libraries were prepared with the TruSeq Stranded mRNA Sample Prep Kit (Illumina) and RNA sequencing was performed in a HiSeq2500 platform (Illumina). Quality control metrics were obtained with FastQC 0.11.7 software (Babraham Bioinformatics). To quantify transcript abundances using the Kallisto 0.43.0 software (Patcher Lab), read counts were aligned to the GRCm38 mouse reference genome cDNA index (Ensembl rel.67). Gene-level differential expression analysis was carried out with the R statistical package Sleuth 0.30.0 comparing the expression of cells cultured alone vs co-culture. Differentially expressed genes were calculated using the Wald statistical test, correcting for multiple testing with the Benjamin-Hochberg method. The false discovery rate (FDR) threshold was set at 1% (q-values <0.01). Ensembl gene transcripts were annotated to Entrez IDs and official gene symbols with the R statistical package BioMart 2.40.3. To normalise for sequencing depth and gene length, transcripts per million (TPM) expression values were calculated. Gene Set Enrichment Analysis (GSEA) was performed with the R statistical package fgsea 1.10.0. The collection of hallmark gene sets from the Molecular Signature Database was used for the GSEA, setting the FDR threshold at 5%. Data was deposited in GEO (GSE163478).

scRNA-seq data generated by *Baryawno et al., 2019* can be found in GEO (GSE128423, https://www.ncbi.nlm.nih.gov/geo/query/acc.cgi?acc=GSE128423).

## Acknowledgements

N Vilaplana-Lopera, G Papatzikas, A Cunningham, and A Erdem were supported by the EU grant HaemMetabolome H2020-MSCA-ITN-2015–675790. U Günther, P Garcia, J J Schuringa, J-B Cazier, and F Schnütgen acknowledge support from the European Commission (HaemMetabolome [EC-675790]). This work was further supported by the Deutsche Forschungsgemeinschaft (DFG, German Research foundation) SFB815, TP A10 (FS). We also acknowledge the Wellcome Trust for supporting access to NMR instruments at the Henry Wellcome Building for Biomolecular NMR in Birmingham (grant number 208400/Z/17/Z). The mouse work was supported by a joint meeting grant of the Northern Norway Regional Health Authority and UiT (Strategisk-HN06-14) to L Arranz. We thank A Villatoro and LM Gonzalez for technical assistance.

## Additional information

### Funding

| Funder | Grant reference number | Author |
| --- | --- | --- |
| Horizon 2020 Framework Programme | H2020-MSCA-ITN-2015-675790 | Grigorios Papatzikas |
| European Commission | HaemMetabolome [EC-675790] | Frank Schnütgen |
| Deutsche Forschungsgemeinschaft | SFB815 TP A10 | Frank Schnütgen |
| Wellcome Trust | 208400/Z/17/Z | Ulrich L Günther |

| Funder | Grant reference number | Author |
| --- | --- | --- |
| Helse Nord RHF | Strategisk-HN06-14 | Lorena Arranz |
| University of Birmingham | 67262-DIF Post-Covid Support Fund | Paloma Garcia |
| European Commission | HaemMetabolome [EC-675790] | Jean-Baptiste Cazier |

The funders had no role in study design, data collection and interpretation, or the decision to submit the work for publication. For the purpose of Open Access, the authors have applied a CC BY public copyright license to any Author Accepted Manuscript version arising from this submission.

## Author contributions

Nuria Vilaplana-Lopera, Conceived and performed the experiments, Conceptualization, Data curation, Formal analysis, Writing – original draft, Writing – review and editing; Vincent Cuminetti, Performed and analysed in vivo experiments, Contributed to interpretation of the data., Writing – review and editing, Investigation, Methodology; Ruba Almaghrabi, Writing – review and editing, Performed experiments; Grigorios Papatzikas, Formal analysis, Writing – review and editing, RNA-seq analysis; Ashok Kumar Rout, Performed NMR experiments and analysis; Mark Jeeves, Methodology, Writing – review and editing; Elena González, Writing – review and editing, Performed experiments; Yara Alyahyawi, Performed experiments; Alan Cunningham, Writing – review and editing, Performed experiments; Ayşegül Erdem, Writing – review and editing, Performed experiments; Frank Schnütgen, Writing – review and editing, Performed experiments; Manoj Raghavan, Resources; Sandeep Potluri, Resources, Writing – review and editing; Jean-Baptiste Cazier, Helped with experimental design; Jan Jacob Schuringa, Resources, Writing – review and editing, Helped with the experimental design; Michelle AC Reed, Writing – review and editing, Performed experiments; Lorena Arranz, Resources, Designed, performed and analysed in vivo experiments, Contributed to interpretation of the data, Critical discussion, Writing – review and editing, Investigation, Supervision, Funding acquisition, Methodology; Ulrich L Günther, Conceptualization, Software, Supervision, Funding acquisition, Methodology, Project administration, Writing – review and editing; Paloma Garcia, Conceptualization, Resources, Supervision, Funding acquisition, Investigation, Methodology, Writing – original draft, Project administration, Writing – review and editing

## Author ORCIDs

Nuria Vilaplana-Lopera ![ORCID] http://orcid.org/0000-0002-3156-9908
Vincent Cuminetti ![ORCID] http://orcid.org/0000-0001-8396-710X
Grigorios Papatzikas ![ORCID] http://orcid.org/0000-0002-0163-4174
Mark Jeeves ![ORCID] http://orcid.org/0000-0001-9736-0990
Jan Jacob Schuringa ![ORCID] http://orcid.org/0000-0001-8452-8555
Lorena Arranz ![ORCID] http://orcid.org/0000-0002-5896-4238
Ulrich L Günther ![ORCID] http://orcid.org/0000-0001-9840-5943
Paloma Garcia ![ORCID] http://orcid.org/0000-0001-5582-8575

## Ethics

AML and PBMC primary specimens' procedures were obtained in accordance with the Declaration of Helsinki at the University Medical Center Groningen, approved by the UMCG Medical Ethical Committee or at the University Hospital Birmingham NHS Foundation Trust, approved by the West Midlands - Solihull Research Ethics Committee (10/H1206//58).

Animal experiments were conducted with the ethical approval of the Norwegian Food and Safety Authority under project number 19472 at the University of Oslo and project number 24739 at UiT - the Arctic University of Norway, with a particular focus on reduction and refinement. Animals were housed under specific opportunistic and pathogen free environment at the Section of Comparative Medicine at the University of Oslo, Norway, and the Unit of Comparative Medicine at UiT - the Arctic University of Norway. The animals were euthanized by $CO_2$ and absence of reflexes was confirmed before necropsy.

## Decision letter and Author response

Decision letter https://doi.org/10.7554/eLife.75908.sa1

Author response https://doi.org/10.7554/eLife.75908.sa2

## Additional files

### Supplementary files
• Supplementary file 1. Primary AML samples' additional information. Table shows information regarding the type of AML, karyotype and additional mutation and risk of the different AML patient samples used during this study.

• MDAR checklist

### Data availability
RNA-seq data has been deposited in GEO under accession number GSE163478. All data generated or analysed during this study are included in the manuscript and supporting file; Source Data files have been provided for all figures. Information about AML patient samples obtained from Martini Hospital (UMCG) (Netherlands) and University Hospital Birmingham NHS Foundation Trust, University of Birmingham (UK) have been provided in Supplementary file 1. Source of mice used can be found in Material and methods.

The following dataset was generated:

| Author(s) | Year | Dataset title | Dataset URL | Database and Identifier |
|---|---|---|---|---|
| Vilaplana-Lopera N | 2021 | Crosstalk between AML and stromal cells triggers acetate secretion through the metabolic rewiring of stromal cells | https://www.ncbi.nlm.nih.gov/geo/query/acc.cgi?acc=GSE163478 | NCBI Gene Expression Omnibus, GSE163478 |

The following previously published dataset was used:

| Author(s) | Year | Dataset title | Dataset URL | Database and Identifier |
|---|---|---|---|---|
| Baryawno N, Przybylski D | 2019 | A cellular taxonomy of the bone marrow stroma in homeostasis and leukemia demonstrates cancer-crosstalk with stroma to impair normal tissue function | https://www.ncbi.nlm.nih.gov/geo/query/acc.cgi?acc=GSE128423 | NCBI Gene Expression Omnibus, GSE128423 |

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
