## [Editor Report]

This article will be of interest to those working in the fields of hematopoiesis, leukemia and cancer microenvironment. The work describes a novel phenomenon whereby the direct interaction of acute myeloid leukemia (AML) cell lines and bone marrow stromal cell lines results in increased production of extracellular acetate in stromal cells, which can then be metabolised by the AML cells. Overall, the data are very thought-provoking but future work some of which is technically challenging, will be necessary for a full appreciation of the biological relevance of this interaction.

---

## [Decision Letter]

**Decision letter after peer review:**

Thank you for submitting your article "Crosstalk between AML and stromal cells triggers acetate secretion through the metabolic rewiring of stromal cells" for consideration by *eLife*. Your article has been reviewed by 3 peer reviewers, and the evaluation has been overseen by a Reviewing Editor and Utpal Banerjee as the Senior Editor. The following individual involved in review of your submission has agreed to reveal their identity: Mick Milson (Reviewer #2).

The suggested revisions are fairly extensive, and due to the current situation with the pandemic, if you will need more time than the usual 2 month time limit for revisions, please let our editorial office know and we will be flexible in adjusting to a reasonable time scale. There is only one major revision allowed, so please take your time (within reason), and attend to all the points raised. At *eLife*, our "no scoop policy" will protect your work if another similar paper appears in press during your revision.

Essential revisions:

1) The conclusion that acetate is used as biofuel by AML cells is not supported by the data at least in the co-culture. Please strengthen this conclusion based on the following comments, and specific comments from Reviewer 2 and 3. The intracellular labelling in Figure 2E does not show a significant increase in any of the TCA cycle intermediates labelling in the SKM-1 cells compared to MS-5 and S3B does not show the MS-5 levels at all. The big increase in acetyl-carnitine without corresponding increase in TCA cycle intermediates makes one wonder if there is a blockage in oxidation of this product.

2) It is important to show that the acetate used by AML cells is produced by MS5 cells. Is it possible to demonstrate not just AML cells uptake of acetate but the full pathway of U-C13 entering MS5 cells and subsequently AML cells? It should be possible to use data from the experiment with U-C13 glucose labelling, to show intracellular labelling in MS-5 and AML cells. These data will be very useful to 1. Confirm glycolysis is upregulated in MS-5 cells in co-culture, 2. Acetate is coming from the labelled pyruvate in MS-5 cells. These data should be presented to support the claims that at the moment is mostly indirect re MS-5 upregulating glycolysis and producing acetate via pyruvate through a non-enzymatic reaction. Please consider specific points from Reviewer 2 and 3 linked to this.

3) The biological relevance of the presented findings needs strengthening, especially given that the AML cells do not grow more on MS5 cells than in single culture. Are the findings linked to AML growth, or perhaps chemoresistance or leukaemia propagation? Please note points from Reviewer 1 and 3 when addressing this.

4) the link between gap junctions, ROS transfer and induction of acetate production needs strengthening. See Reviewr 2 point: the experiments are overall convincing including re the role of gap junctions. What is missing is direct proof that inhibiting gap junction would reduce acetate production in the co-culture and this is something that needs done to support the statement that it is the ROS transfer via gap junction that is leading to acetate production. And Reviewer 1: is the metabolic rewiring and overexpression of gap junctions a consequence of increased ROS? Or are gap junctions upregulated first, leading to increased ROS intake and metabolic re-wiring?

5)While further extensive in vivo experiments may be beyond the scope of this work and would require considerably longer than 3 months to complete, is there any evidence from already published transcriptomic datasets that stroma cells in vivo upregulate the same genes/pathways? Is it possible to identify the specific cell type responsible for the mechanism presented here?

6) Please clarify the rational behind the choice of AML cell lines and human healthy control samples used.

Further points:

7) Please consider specific points from each reviewer in your response letter.

*Reviewer #1 (Recommendations for the authors):*

1) Is the metabolic rewiring and overexpression of gap junctions a consequence of increased ROS? Or are gap junctions upregulated first, leading to increased ROS intake and metabolic re-wiring?

2) Is there evidence from in vivo studies that stroma cells upregulate the same genes/pathways? Is it possible to identify the specific cell type responsible for the mechanism presented here? If a specific cell type appears to be important with this, it would be exciting to see an effect following specific targeting of that cell type, although this may be beyond the scope of the presented work.

3) Is the mechanism presented here supporting AML growth prior or also following chemotherapy administration?

*Reviewer #2 (Recommendations for the authors):*

The manuscript by Vilaplana-Lopera et al., highlights a novel interesting metabolic cross-talk between the stroma and AML cells through production of acetate. The main conclusion of the paper, ie that acetate production is increased in co-culture most likely because of stroma production and is taken up by AML cells, is supported by the data. Some of the other conclusions re the utilisation of acetate in AML cells and the exact mechanism leading to acetate production might require further experimental work to be fully supported. The authors make an effort to validate their findings in primary AML cells model and in vivo although the mechanistic insight is mostly done in AML cell lines grown on a MS-5 stromal layer. As a result whether the described interaction is also present in other stroma/AML combination remains to be demonstrated. However I think this is not necessary for this paper.

Technically the paper appears rigorous although I would not be able to comment on the technical aspects of the NMR analysis and I would hope some of the other reviewers can comment on that.

Specific comments:

Figure 1 – overall I find the data here convincing. My main criticism is that the author appear to support the notion that acetate comes from MS-5 cells only while it cannot be excluded that AML cells might be reprogrammed by the stroma to a certain extent to produce acetate as in 1D for SKM1 and HL60 there is a slight increase in acetate following interruption of the coculture. See below on how to strengthen this conclusion or otherwise reword the discussion.

Figure 2 – what do the author mean by physiological conditions? How is the acetate concentration of 0.25mM considered physiological and are they implying that their co-culture system is not physiological? Also the conclusion that acetate is used as biofuel by AML cells is not supported by the data at least in the coculture. The intracellular labelling in Figure 2E does not show a significant increase in any of the TCA cycle intermediates labelling in the SKM-1 cells compared to MS-5 and S3B does not show the MS-5 levels at all. Infact the big increase in acetyl-carnitine without corresponding increase in TCA cycle intermediates makes one wonder if there is a blockage in oxidation of this product.

One way to address that would be to measure ATP in AML cells or oxygen consumption following co-culture. Alternative fates for acetate, i.e. lipid biosynthesis, might otherwise need to be explored.

Figure 3 – Upregulation of glycolysis in MS-5 in coculture is indirectly demonstrated from transcriptomic data. There is no direct proof of it and infact based on the patterns of glucose and lactate levels in coculture shown in S1, the authors themselves had concluded that "the overall increase in glycolysis in co-culture was just a result of culturing both cell types together". As the authors have done U-C13 glucose labelling , they might have data for intracellular labelling in MS-5 and AML cells. These data will be very useful to 1. Confirm glycolysis is upregulated in MS-5 cells in co-culture, 2. Acetate is coming from the labelled pyruvate in MS-5 cells. These data should be presented to support the claims that at the moment is mostly indirect re MS-5 upregulating glycolysis and producing acetate via pyruvate through a non-enzymatic reaction.

Figure 4 – Regarding production of acetate. Overall the data are supportive but do not provide conclusive evidence that acetate production is via non-enzymatic reaction. The authors could inhibit pyruvate carrier to see if that has an effect on acetate production (prediction would be not).The intracellular labelling from U-C13 glucose proposed above would be very helpful to clarify this too. I cannot fully interpret their thiamine depletion experiment as it appears to have been done not in co-culture system so I am not sure how much it does support that no production of acetate via enzymatic happens in the co-culture system.

Finally treatment with HDAC might help ensure that indeed the acetate is mostly coming via production through glycolysis although this is supported by their labelling and might not be a key experiment.

With regards to the role of ROS, the experiments are overall convincing including re the role of gap junctions. What is missing is direct proof that inhibiting gap junction would reduce acetate production in the co-culture and this is something that needs done to support the statement that it is the ROS transfer via gap junction that is leading to acetate production.

Discussion – the authors state that acetate is used to feed TCA cycle to generate energy. This is not supported by the data as labelling of TCA cycle is unclear from co-culture and ATP production or oxygen consumption not shown. Acetate fate can be complex including in lipid biosynthesis or to support TCA cycle intermediates for biosynthesis of other macromolecules (cataplerosis). As such the authors should be careful about drawing conclusions which are not supported by their data and provide further evidence to their claims or change discussion (see above).

Also the conclusion “Mechanistically, our data revealed that acetate secretion involves not only a higher glycolytic rate but also the non-enzymatic ROS-mediated conversion of pyruvate to acetate, as cells grown in thiamine-free media were still capable to produce acetate” needs further experimental support although their data suggest this is likely true. The experiment suggested above to support this statement would also further support the conclusion that acetate comes mostly from the stroma.

The conclusion "We showed that AML cells are capable of modulating the metabolism of stromal cells by transferring ROS via gap junctions resulting in an increased secretion of acetate and its subsequent accumulation in the extracellular medium" is not fully supported as it is not shown that inhibiting gap junction reduces acetate.

Also "Furthermore, we found that AML cells consume the secreted acetate and use it as an energy source, by fluxing it into the TCA cycle" needs to be reconsidered as per above.

*Reviewer #3 (Recommendations for the authors):*

In general, I found the data in the manuscript to be novel, interesting and thought provoking. However, I would like to suggest some experiments that would help to solidify the claims that are made in the manuscript as well as potentially increase the impact of the work.

1) While the acetate secretion by stromal cells and uptake by AML cells is a very interesting finding, I'm sure that many readers would join me in asking: what is the biological relevance of this? Is it simply a metabolic quirk or does it convey some advantage to the AML cells? Can the authors find any aspect of altered biology in the AML cells that may be relevant to disease progression, therapy resistance, relapse etc? Is it possible to block/inhibit AML import of acetate or stromal cell secretion of acetate in order to assess the impact of this phenomenon? Can alterations in culture conditions or treatment with AML therapy relevant drugs demonstrate a dependency/advantage conferred by stromal derived acetate? If so, this would significantly boost the impact of the work and demonstrate the importance of the mechanism.

2) It may seem to be nit-picking, but I noted that it has not been directly shown that AML cells can metabolise the acetate that is produced by stromal cells upon co-culture. The metabolic tracing experiments have been performed using labelled acetate that has been added to the culture medium. Since this is a central claim of the manuscript, I think it is important to show this, if at all possible. Can the stromal cells be grown in media that will label the carbon molecules in acetate prior to co-culture, as opposed to adding acetate directly to the culture medium?

3) Although the authors show no effect of AML cell growth upon co-culture, they do not assess whether the growth of stromal cells is altered. This would have important implications for the interpretation of any alteration in metabolism of these cells.

4) I would strongly encourage the authors to use an additional method to rescue ROS production rather than NAC, which is problematic in this experimental setting since it can directly feed into the metabolic pathways that are being studied. As well as considering the use of other chemicals that are considered to act as antioxidants, I would also encourage the authors to explore overexpression of ROS detoxifying enzymes such as superoxide dismutase and catalase. By taking the latter approach, one could specifically detoxify ROS in either AML cells or stromal cells, which would help to strengthen the proposed transfer mechanism. Genetic ablation/knockdown of the same enzymes could also be used as an elegant way to strengthen the ROS transfer concept.

5) Relating to 4), I was very surprised that the authors referenced the prior work documenting transfer of mitochondria between stromal cells and HSCs via gap junctions, but did not address whether this was of any mechanistic relevance to their own observations. Since mitochondria are thought to be the major source of intracellular ROS, this would seem to be a clear alternative possible mechanism which would yield similar experimental outcomes to those observed by the authors.

6) The authors should make sure that the primary patient AML co-culture experiments have been conducted using identical culture conditions. It is problematic for the reader to assess the experimental outcome if culture conditions have been inconsistently altered across different test and control samples. I appreciate that such patient samples are not easy to acquire or work with, but the authors should repeat these experiments if necessary.

7) Related to 6), could the authors clearly state what the primary hematopoietic control cells that they use in the control experiments are? In the figure, it seems to suggest that peripheral blood mononuclear cells were used, but in the methods, it seems to suggest that CD34+ cells are used, which seems inconsistent with the description of PBMNCs. If the authors used CD34+ cells from AML patients, then I would suggest that CD34+ cells from the bone marrow of normal donors is probably the best control for this experiment.

My final comment is beyond the scope of the current manuscript, but I would also encourage the authors to pursue this line of investigation further using the murine MLL-AF9 model that they perform a preliminary characterization with in this manuscript. This would afford the possibility of using a wide range of genetic models to interrogate mechanism in vivo, as well as a being an attractive model to explore biological relevance in the context of disease evolution and therapy.

---

## [Author Response]

Essential revisions:1) The conclusion that acetate is used as biofuel by AML cells is not supported by the data at least in the co-culture. Please strengthen this conclusion based on the following comments, and specific comments from Reviewer 2 and 3. The intracellular labelling in Figure 2E does not show a significant increase in any of the TCA cycle intermediates labelling in the SKM-1 cells compared to MS-5 and S3B does not show the MS-5 levels at all. The big increase in acetyl-carnitine without corresponding increase in TCA cycle intermediates makes one wonder if there is a blockage in oxidation of this product.

We have taken into consideration the reviewers’ comments and performed intracellular labelling by adding [2-^13^C] acetate to the coculture before harvesting and separating AML and MS-5 cells 30 minutes after acetate addition. The new data clearly shows labelling in TCA intermediates: citrate, glutamate, oxoglutarate, malate, proline and succinate in SKM-1 and Kasumi-1 cells whilst labelled citrate, glutamate and succinate are the main metabolites found in HL-60 cells. In contrast, no labelled metabolites were found in the stromal cells. We have included this data in Figure 2C and Figure 2-Supplement Figure 2B-C (left panels). In this respect the result is the same as for MS-5 and SKM-1 grown in isolation.

We have also performed NMR experiments to detect intracellular labelling in MS-5 at 8 hours to complete former figure S3B (new Figure 2-Supplement Figure 2B-C, right panels). Consistent with observations for MS-5 in co-culture with SKM1 at 8h, no intracellular labelling was detected in MS-5 at any given time when co-cultured with any of the three AML cell lines. As the MS-5 labelling experiments did not show any labelling in TCA intermediates, neither at 2h nor at 8h (n=3), the intracellular analysis of MS5 at 30 minutes labelling and 8h labelling for Kasumi-1 and HL-60 was performed only in one set to confirm that no label was incorporated as previously agreed with the reviewer editor.

We have also explored other possible products of acetate by analysing the intracellular non-polar fraction of AML cells after labelling with [2-^13^C] acetate for 8 and 48 hours. Our data revealed that at 8h, no label incorporation could be detected in the non-polar or lipid fraction (New Figure 2- Supplement Figure 3). In contrast, at 48h, the ^1^H spectrum shows ^13^C-coupled ^1^H-signals at 0.75-0.80 ppm arising from ^13^C incorporation (^13^C ‘satellite’ signals) in lipid CH_3_ groups (New Figure 2D and Figure 2-Supplement Figure 4). While NMR cannot determine the chain length of the lipids, integration of the signals yields the overall label incorporation in any lipid CH_3_ groups, indicating that acetate is being used for lipid biosynthesis.

With regards to the big increase in labelled acetyl carnitine, we would like to clarify that the graphs represent percentage of label incorporation from labelled acetate and not metabolite concentration. What we can conclude from our data is that the label incorporation in most of the acetyl moiety of acetycarnitine (C9) is higher, hence, highlighting that it derives from acetate.

It is possible that an increase in the concentration of acetylcarnitine is also happening. This possibility would be in line with other reports showing that when formation of acetyl-CoA exceeds use by the TCA cycle, an increase in acetylcarnitine is observed which could function as a reservoir to keep the mitochondrial acetyl-CoA/free CoA ratio low to sustain TCA cycle flux (Lindeboom *et al.,* JCI, 2014). Therefore, the addition of labelled acetate could result in a big increase of labelled acetylcarnitine as storage to sustain TCA flux as previously reported. A block in b-oxidation would be supported by the slow incorporation into lipids but there is insufficient data to proof this point. However, high levels of label incorporation in acetylcarnitine and other TCA cycle intermediates are in agreement with the proposed mechanism where AML cells import and use acetate to feed the TCA cycle and for lipogenesis.

2) It is important to show that the acetate used by AML cells is produced by MS5 cells. Is it possible to demonstrate not just AML cells uptake of acetate but the full pathway of U-C13 entering MS5 cells and subsequently AML cells? It should be possible to use data from the experiment with U-C13 glucose labelling , to show intracellular labelling in MS-5 and AML cells. These data will be very useful to 1. Confirm glycolysis is upregulated in MS-5 cells in co-culture, 2. Acetate is coming from the labelled pyruvate in MS-5 cells. These data should be presented to support the claims that at the moment is mostly indirect re MS-5 upregulating glycolysis and producing acetate via pyruvate through a non-enzymatic reaction. Please consider specific points from Reviewer 2 and 3 linked to this.

We agree with this concern raised by the reviewers as we also asked ourselves the same question. In part this question is answered in the question above, although it would be nicer to proof flux from MS-5 to AML cells directly. Unfortunately, this is extremely difficult, if not impossible. We believe this is because acetate is not transferred via any specific junction, but rather secreted from MS-5 cells (as proven by our data) and taken up by AML cells.

Nevertheless, we have attempted to label MS-5 cells with [U-^13^C]glucose prior to co-culture with AML cells. However, after removing the spent medium containing labelled acetate (and the remaining labelled glucose before co-culturing cells), the amount of label left in MS-5 cells is not sufficient to accumulate in the medium and be taken up by AML cells. Moreover, as new non-labelled glucose is added in the form of fresh medium when starting the co-culture, any acetate newly generated from glucose is unlabelled, hence decreasing even more the percentage of labelled acetate. This is consistent with the view described above that MS-5 cells actually do not store acetate, but rather secrete it into the medium, and part of it is subsequently taken up by AML cells. Without a sufficient amount of labelled acetate present in the medium it is impossible to detect labelling in AML cells by NMR (and possibly by any other technique).

We have performed 2D HSQCs spectra on intracellular polar metabolites in MS-5 cells cultured alone vs in co-culture labelled with [1,2-^13^C]glucose and have found that labelling in acetate, alanine and lactate is higher in co-culture (although only significant for lactate). This is more clear when %13C are represented as ratios.

Interestingly, we do not observe pyruvate intracellularly in MS-5 cells and therefore cannot assess label incorporation, probably owing to fast turn-over into acetate.

However, we believe that the build-up of acetate in the medium and the general increase in label incorporation in acetate and other metabolites derived from this pathway supports our claims on the upregulation of glycolysis in MS-5 cells in coculture and MS-5 cells as the major source for acetate. This data has been incorporated in Figure 3E.

3) The biological relevance of the presented findings needs strengthening, especially given that the AML cells do not grow more on MS5 cells than in single culture. Are the findings linked to AML growth, or perhaps chemoresistance or leukaemia propagation? Please note points from Reviewer 1 and 3 when addressing this.

To understand the biological relevance of our findings we have performed survival assays when cells are treated with the chemotherapeutic drug cytarabine (AraC) and with the ACSS2 inhibitor (ACSS2i) alone or in combination, in medium without glucose in the presence of 4mM acetate or in normal media media (containing glucose but not acetate). This experiment was not performed in coculture as we reasoned it would be clearer to use just the cancer cells, to avoid the possibility of other metabolites provided by MS-5 having an effect on cell survival.

Our data shows that the AraC treatment affects cell survival in a cell line-dependent manner, with HL-60 being the more sensitive to this drug then Kasumi-1 and SKM-1. Interestingly and supporting the importance of acetate in cell metabolism, the survival of the cells treated with ACSS2i is greatly compromised and the combination of AraC and ACSS2i showed very consistently sensitisation of all the AML cell lines studied to AraC. We also performed these experiments in medium supplemented with a high concentration of acetate (4mM) and found a partial recovery of this sensitisation. We believe that these results highlight the importance of acetate for AML cells and reveal a potential role of acetate in AML chemoresistance. This data can be found in the new Figure 2E and explained in the text in lines 199-220.

Additionally, we have performed in vivo experiments in which mice were transplanted with leukaemic cells and injected with a ROS inducer (TBHP) alone or in combination with the gap junction inhibitor CBX. TBHP injection accelerated the development of leukaemia, promoting an increase in the total monocyte count in peripheral blood and reducing the lifespan of TBHP- versus vehicle-treated leukaemic mice. Moreover, our data showed that administration of the gap junction inhibitor partially counteracts the effect of TBPH in both monocyte counts and survival. This data can be found in the new Figure 5G and 5H and explained in the text in lines 357-367. These in vivo results support a biological relevance in leukaemia progression for our observations in vitro involving ROS transfer through gap junctions between AML and stromal cells.

4) the link between gap junctions, ROS transfer and induction of acetate production needs strengthening. See Reviewr 2 point: the experiments are overall convincing including re the role of gap junctions. What is missing is direct proof that inhibiting gap junction would reduce acetate production in the co-culture and this is something that needs done to support the statement that it is the ROS transfer via gap junction that is leading to acetate production.

As suggested by the reviewer and provide with a direct proof that inhibiting gap junctions reduces acetate production, we have performed co-cultures with CBX to inhibit the gap junctions and measured acetate extracellular levels by NMR. Our results show that gap junction inhibition reduces acetate production in co-cultures in the three cell lines. The results can be found in new Figure 5F and explained in the text in lines 349-356.

And Reviewer 1: is the metabolic rewiring and overexpression of gap junctions a consequence of increased ROS? Or are gap junctions upregulated first, leading to increased ROS intake and metabolic re-wiring?

This is an interesting point that we have addressed by performing experiments using a ROS-scavenging enzyme (catalase) as suggested by reviewer 3. First, we performed a dose curve and observed that a decrease in ROS levels could be seen at concentrations above 100ug/ml. We chose 500ug/ml of catalase to perform further experiments. We pre-treated the AML cells with catalase for 24h and assessed gap junction formation by following the calcein-AM transfer protocol as before. Our results showed that catalase treatment reduces calcein-AM transfer in co-culture and, thus, the ability of forming gap junctions (new Figure 5I). This result together with new NMR data showing that in the presence of GAP junction inhibitor (CBX) the production of acetate is reduced (Figure 5F) provides clarification on the order of events, which we believe occurs as follows: high ROS in AML cells induces gap junction formation as a mechanism of ROS transfer to stromal cells causing the metabolic rewiring (model in new Figure 6).

5) While further extensive in vivo experiments may be beyond the scope of this work and would require considerably longer than 3 months to complete, is there any evidence from already published transcriptomic datasets that stroma cells in vivo upregulate the same genes/pathways? Is it possible to identify the specific cell type responsible for the mechanism presented here?

As suggested by the reviewer we have made use of published data sets in particular the single cell RNA seq data published in Cell by the group of David Scadden (Baryawno et al., Cell 2019) in which they analysed the transcriptome of bone marrow stroma in homeostasis versus leukaemia (using MLL-AF9 knock-in mice: Corral et al., 1996). These data sets show three specific clusters (cluster 1, 4 and 7) present in leukaemic stroma, which have upregulated the glycolysis and hypoxia pathways. The proportion of osteoprogenitor cluster (cluster 7) was shown to increase in the stroma of leukaemic mice, and cells in this cluster shared the same hallmark gene sets that MS-5 in co-culture: TNFalpha, Hypoxia, allograft rejection, glycolysis, IFNγ, K-ras, complement, epithelial to mesenchymal transition, IL2-STAT5 etc (Table S4 in Baryawno et al., 2019).

This information has been included in the discussion, in lines 429-435.

6) Please clarify the rational behind the choice of AML cell lines and human healthy control samples used.

We made use of three different AML cell lines during this work to cover a range of AML and not looking at a specific subtype. The subtypes chosen would reflect M2 immature AML (HL-60), MS-5 mature AML (SKM-1) and AML with the common translocation AML1-ETO (Kasumi-1). As requested by the reviewer we have included this information in line 88.

Human healthy control samples were CD34+ cells from the peripheral blood. This information has been included in the Materials and methods section, line 508 and 517.

Reviewer #1 (Recommendations for the authors):1) Is the metabolic rewiring and overexpression of gap junctions a consequence of increased ROS? Or are gap junctions upregulated first, leading to increased ROS intake and metabolic re-wiring?

Please, see answer to point 4 in “essentials”.

2) Is there evidence from in vivo studies that stroma cells upregulate the same genes/pathways? Is it possible to identify the specific cell type responsible for the mechanism presented here? If a specific cell type appears to be important with this, it would be exciting to see an effect following specific targeting of that cell type, although this may be beyond the scope of the presented work.

Please, see answer to point 5 in “essentials”.

3) Is the mechanism presented here supporting AML growth prior or also following chemotherapy administration?

Please, see answer to point 3 in “essentials”.

Reviewer #2 (Recommendations for the authors):The manuscript by Vilaplana-Lopera et al., highlights a novel interesting metabolic cross-talk between the stroma and AML cells through production of acetate. The main conclusion of the paper, ie that acetate production is increased in co-culture most likely because of stroma production and is taken up by AML cells, is supported by the data. Some of the other conclusions re the utilisation of acetate in AML cells and the exact mechanism leading to acetate production might require further experimental work to be fully supported. The authors make an effort to validate their findings in primary AML cells model and in vivo although the mechanistic insight is mostly done in AML cell lines grown on a MS-5 stromal layer. As a result whether the described interaction is also present in other stroma/AML combination remains to be demonstrated. However I think this is not necessary for this paper.Technically the paper appears rigorous although I would not be able to comment on the technical aspects of the NMR analysis and I would hope some of the other reviewers can comment on that.Specific comments:Figure 1 – overall I find the data here convincing. My main criticism is that the author appear to support the notion that acetate comes from MS-5 cells only while it cannot be excluded that AML cells might be reprogrammed by the stroma to a certain extent to produce acetate as in 1D for SKM1 and HL60 there is a slight increase in acetate following interruption of the coculture. See below on how to strengthen this conclusion or otherwise reword the discussion

As suggested by the reviewer, the discussion has been reworded to contemplate the possibility of AML metabolism being rewired. This can be found in line 410-412.

Figure 2 – what do the author mean by physiological conditions? How is the acetate concentration of 0.25mM considered physiological and are they implying that their co-culture system is not physiological?

By physiological conditions we mean normal levels in plasma, we have changed the word in the text to make it clearer (line 166 and 168). A reference supporting that normal acetate levels in human plasma are 0.25mM has also been included (Gao *et al.,* Nature comms 2016; reference number 26).

Also the conclusion that acetate is used as biofuel by AML cells is not supported by the data at least in the coculture. The intracellular labelling in Figure 2E does not show a significant increase in any of the TCA cycle intermediates labelling in the SKM-1 cells compared to MS-5 and S3B does not show the MS-5 levels at all. Infact the big increase in acetyl-carnitine without corresponding increase in TCA cycle intermediates makes one wonder if there is a blockage in oxidation of this product.One way to address that would be to measure ATP in AML cells or oxygen consumption following co-culture. Alternative fates for acetate, i.e. lipid biosynthesis, might otherwise need to be explored

Please, see answer to point 1 in “essentials”.

Figure 3 – Upregulation of glycolysis in MS-5 in coculture is indirectly demonstrated from transcriptomic data. There is no direct proof of it and infact based on the patterns of glucose and lactate levels in coculture shown in S1, the authors themselves had concluded that “the overall increase in glycolysis in co-culture was just a result of culturing both cell types together”. As the authors have done U-C13 glucose labelling , they might have data for intracellular labelling in MS-5 and AML cells. These data will be very useful to 1. Confirm glycolysis is upregulated in MS-5 cells in co-culture, 2. Acetate is coming from the labelled pyruvate in MS-5 cells. These data should be presented to support the claims that at the moment is mostly indirect re MS-5 upregulating glycolysis and producing acetate via pyruvate through a non-enzymatic reaction.

Please, see answer to point 2 in “essentials”.

Figure 4 – Regarding production of acetate. Overall the data are supportive but do not provide conclusive evidence that acetate production is via non-enzymatic reaction. The authors could inhibit pyruvate carrier to see if that has an effect on acetate production (prediction would be not).The intracellular labelling from U-C13 glucose proposed above would be very helpful to clarify this too. I cannot fully interpret their thiamine depletion experiment as it appears to have been done not in co-culture system so I am not sure how much it does support that no production of acetate via enzymatic happens in the co-culture system.

The reaction doesn’t take place in the mitochondria, so we don’t think that the inhibition of the pyruvate carrier would affect acetate production (please see article published by the group of Locasale reference: Liu et al., Cell 2018).

The experiment with thiamine free media aimed to support our previous publication showing that it is possible to produce acetate by a non-enzymatic reaction in the presence of ROS (adding H2O2). Our previous publication described this reaction in a cell-free system. We felt that we needed to prove that this reaction occurs in a cellular system. If the reviewer thinks that our results are confusing we would be happy to remove them.

Finally treatment with HDAC might help ensure that indeed the acetate is mostly coming via production through glycolysis although this is supported by their labelling and might not be a key experiment.

We thank the reviewer for this suggestion. As the reviewer acknowledged, our labelling experiments already supports that acetate is produced through glycolysis and thus we decided to focus our efforts on answering the essential questions highlighted by the editor.

With regards to the role of ROS, the experiments are overall convincing including re the role of gap junctions. What is missing is direct proof that inhibiting gap junction would reduce acetate production in the co-culture and this is something that needs done to support the statement that it is the ROS transfer via gap junction that is leading to acetate production.

Please, see answer to point 4 in “essentials”.

Discussion – the authors state that acetate is used to feed TCA cycle to generate energy. This is not supported by the data as labelling of TCA cycle is unclear from co-culture and ATP production or oxygen consumption not shown. Acetate fate can be complex including in lipid biosynthesis or to support TCA cycle intermediates for biosynthesis of other macromolecules (cataplerosis). As such the authors should be careful about drawing conclusions which are not supported by their data and provide further evidence to their claims or change discussion (see above).

We thank the reviewer for this comment. We have removed the conclusion of using the TCA cycle to generate energy as suggested by the reviewer. (Line 394)

Also the conclusion “Mechanistically, our data revealed that acetate secretion involves not only a higher glycolytic rate but also the non-enzymatic ROS-mediated conversion of pyruvate to acetate, as cells grown in thiamine-free media were still capable to produce acetate” needs further experimental support although their data suggest this is likely true. The experiment suggested above to support this statement would also further support the conclusion that acetate comes mostly from the stroma.

As explained above, this experiment was performed to support our work in a cell-free system. Additionally, our labelling experiments and RNA-seq experiments do not show an upregulation of the enzyme that converts acetate from pyruvate. We have changed this conclusion to tone down the message and write “quite likely” instead of “revealed”:

“Mechanistically, our data revealed that acetate secretion involves not only a higher glycolytic rate but also very probably the non-enzymatic ROS-mediated conversion of pyruvate to acetate, as cells grown in thiamine-free media were still capable to produce acetate and expression of keto acid dehydrogenase was not upregulated”. (Line 395).

The conclusion “We showed that AML cells are capable of modulating the metabolism of stromal cells by transferring ROS via gap junctions resulting in an increased secretion of acetate and its subsequent accumulation in the extracellular medium” is not fully supported as it is not shown that inhibiting gap junction reduces acetate.

We have maintained this conclusion based on the new data presented in Figure 5F, in which we show a reduction in acetate extracellular levels when cells are grown in coculture in the presence of the gap junction inhibitor (CBX).

Also “Furthermore, we found that AML cells consume the secreted acetate and use it as an energy source, by fluxing it into the TCA cycle” needs to be reconsidered as per above.

As above, we have also removed “use as energy source”. Line 459

Reviewer #3 (Recommendations for the authors):In general, I found the data in the manuscript to be novel, interesting and thought provoking. However, I would like to suggest some experiments that would help to solidify the claims that are made in the manuscript as well as potentially increase the impact of the work.1) While the acetate secretion by stromal cells and uptake by AML cells is a very interesting finding, I’m sure that many readers would join me in asking: what is the biological relevance of this? Is it simply a metabolic quirk or does it convey some advantage to the AML cells? Can the authors find any aspect of altered biology in the AML cells that may be relevant to disease progression, therapy resistance, relapse etc? Is it possible to block/inhibit AML import of acetate or stromal cell secretion of acetate in order to assess the impact of this phenomenon? Can alterations in culture conditions or treatment with AML therapy relevant drugs demonstrate a dependency/advantage conferred by stromal derived acetate? If so, this would significantly boost the impact of the work and demonstrate the importance of the mechanism.

Please, see answer to point 4 in “essentials”.

2) It may seem to be nit-picking, but I noted that it has not been directly shown that AML cells can metabolise the acetate that is produced by stromal cells upon co-culture. The metabolic tracing experiments have been performed using labelled acetate that has been added to the culture medium. Since this is a central claim of the manuscript, I think it is important to show this, if at all possible. Can the stromal cells be grown in media that will label the carbon molecules in acetate prior to co-culture, as opposed to adding acetate directly to the culture medium?

Please, see answer to point 2 in “essentials”.

3) Although the authors show no effect of AML cell growth upon co-culture, they do not assess whether the growth of stromal cells is altered. This would have important implications for the interpretation of any alteration in metabolism of these cells.

We understand the concerns of the reviewer. Stromal growth could not be altered as stromal cells are grown until confluency prior to the experiment.

4) I would strongly encourage the authors to use an additional method to rescue ROS production rather than NAC, which is problematic in this experimental setting since it can directly feed into the metabolic pathways that are being studied. As well as considering the use of other chemicals that are considered to act as antioxidants, I would also encourage the authors to explore overexpression of ROS detoxifying enzymes such as superoxide dismutase and catalase. By taking the latter approach, one could specifically detoxify ROS in either AML cells or stromal cells, which would help to strengthen the proposed transfer mechanism. Genetic ablation/knockdown of the same enzymes could also be used as an elegant way to strengthen the ROS transfer concept.

We agree with the reviewer that experiments with genetic ablation/knockdown of catalase or superoxide dismutase would be a very good way to strengthen our results. Still, given the amount of time that we required for performing the essential experiments, these experiments have not been pursued.

As suggested by the reviewer, we have performed additional experiments using the ROS-scavenging enzyme catalase. Treating cocultures with catalase affected acetate production by MS-5 (New Figure 4-Supplement Figure 1C).

Additionally, based on the suggestion of using catalase to strengthen the proposed mechanism, we have performed experiments to determine the order of events (ROS/gap junction) and found a reduction in gap junction formation measured by calcein transfer when co-cultures are treated with catalase (New Figure 5I). Please, see answer to point 4 in “essentials”.

5) Relating to 4), I was very surprised that the authors referenced the prior work documenting transfer of mitochondria between stromal cells and HSCs via gap junctions, but did not address whether this was of any mechanistic relevance to their own observations. Since mitochondria are thought to be the major source of intracellular ROS, this would seem to be a clear alternative possible mechanism which would yield similar experimental outcomes to those observed by the authors.

As mentioned by the reviewer, we were aware and cited the work of others revealing the importance of mitochondria and activation of glutathione-related antioxidant pathways in AML as a way of detox from ROS excess due to chemotherapy (Forte *et al.,* Cell Metabolism,2020 reference 14). Although the work from Forte *et al.,* did not explore the mechanism of mitochondria transfer, previous work by others have linked the mitochondrial transfer to the formation of tunnelling nanotubes (TNT) (Vignais et al., Stem cells International 2017; Kolba et al., Cell Death and Dis, 2019, Moschoi et al., Blood 2016). To our knowledge, mitochondria transfer between AML and stromal cells has never been linked directly to gap junctions Although transference of mitochondria through gap junction may indeed occur, our work aims to highlight a novel mechanism of ROS transfer (ROS/Gap/metabolic rewiring) that the AML cells use beyond mitochondria transfer.

6) The authors should make sure that the primary patient AML co-culture experiments have been conducted using identical culture conditions. It is problematic for the reader to assess the experimental outcome if culture conditions have been inconsistently altered across different test and control samples. I appreciate that such patient samples are not easy to acquire or work with, but the authors should repeat these experiments if necessary.

We understand the concern of the reviewer. This was mainly due to the different growth conditions that were established in the two labs, JJ Schruinga’s lab where the first author of the paper spent a placement and our lab in the University of Birmingham. Still, I would think that the fact that by using two different culture conditions primary AML behave in the same way regarding acetate would be seen as a strength rather than a weakness, as it validates our observations in two different settings. Moreover, data is relative to MS-5 cells cultured in the exact conditions as the matching AML co-cultures.

7) Related to 6), could the authors clearly state what the primary hematopoietic control cells that they use in the control experiments are? In the figure, it seems to suggest that peripheral blood mononuclear cells were used, but in the methods, it seems to suggest that CD34+ cells are used, which seems inconsistent with the description of PBMNCs. If the authors used CD34+ cells from AML patients, then I would suggest that CD34+ cells from the bone marrow of normal donors is probably the best control for this experiment.

We used CD34+ cells from peripheral blood. We have changed PBMC in the figure legend by “healthy” (Figure 1F).

My final comment is beyond the scope of the current manuscript, but I would also encourage the authors to pursue this line of investigation further using the murine MLL-AF9 model that they perform a preliminary characterization with in this manuscript. This would afford the possibility of using a wide range of genetic models to interrogate mechanism in vivo, as well as a being an attractive model to explore biological relevance in the context of disease evolution and therapy.

We thank the reviewer for his comment; indeed this is something that we would like to pursue further.